# A heterocyte glycolipid-based calibration to reconstruct past continental climate change

Thorsten Bauersachs [1]✉, James M. Russell [2], Thomas W. Evans [3], Antje Schwalb [4] &
Lorenz Schwark [1,5]

Understanding Earth's response to climate forcing in the geological past is essential to reliably predict future climate change. The reconstruction of continental climates, however, is hampered by the scarcity of universally applicable temperature proxies. Here, we show that heterocyte glycolipids (HGs) of diazotrophic heterocytous cyanobacteria occur ubiquitously in equatorial East African lakes as well as polar to tropical freshwater environments. The relative abundance of $HG_{26}$ diols and keto-ols, quantified by the heterocyte diol index ($HDI_{26}$), is significantly correlated with surface water temperature (SWT). The first application of the $HDI_{26}$ to a ~37,000 year-long sediment record from Lake Tanganyika provides evidence for a ~4.1 °C warming in tropical East Africa from the last glacial to the beginning of the industrial period. Given the worldwide distribution of HGs in lake sediments, the $HDI_{26}$ may allow reconstructing SWT variations in polar to tropical freshwater environments and thereby quantifying past continental climate change.

[1] Christian-Albrechts-University, Kiel, Germany. [2] Brown University, Providence, RI, USA. [3] Massachusetts Institute of Technology, Cambridge, MA, USA. [4] Technische Universität Braunschweig, Braunschweig, Germany. [5] Curtin University, Bentley, WA, Australia. ✉email: thorsten.bauersachs@ifg.uni-kiel.de

Earth is currently experiencing major alterations of its climate including an increase in surface temperatures not observed in the historic past[1]. Climate models consistently forecast that future global temperatures will increase but they disagree in terms of magnitude, which results in particularly large uncertainties in estimating the extent of future climate change[2]. A means to better constrain future climate change is to study past climate states and transitions using proxy records obtained from climate archives. Organic temperature proxies ($U^{K'}_{37}$, $TEX_{86}$, LDI)[3–5] have proved to be indispensable in this context. These tools are frequently employed in marine sediment sequences to reconstruct past sea surface temperatures and have been used to establish high-resolution palaeotemperature records from all parts of the oceans[5–7] and in sediments dating back to the Early Jurassic (~200 Ma)[8]. As such, organic temperature proxies have significantly broadened our knowledge on Earth's long-term climate evolution over geological timescales[9,10] as well as during abrupt climate change events[11].

Lakes are outstanding archives of past climate change, as they contain continuous, high-resolution sediment records, are widespread, and highly responsive to climate forcing. Obtaining climate-relevant data from lacustrine sediments using lipid palaeothermometers, however, is not always straightforward. The $TEX_{86}$ is only applicable in some large lakes[12] and long-chain alkenones (needed for the calculation of the $U^{K}_{37}$) are generally absent from low-latitude lake sediments[13]. More recently, the $MBT'_{5Me}$ was proposed as a tool to reconstruct past terrestrial temperature[14] but in lakes, its application is complicated due to mixed contributions of aquatic and terrestrially derived branched glycerol dialkyl glycerol tetraethers (GDGTs) with contrasting temperature adaptations[15]. Additional temperature proxies to extract the climate signal stored in lake sediments are thus needed to generate accurate reconstructions of past continental climate change that allow testing climate model hindcasts.

Heterocyte glycolipids (HGs) have recently been proposed as novel tools in reconstructing surface water temperatures (SWTs) in freshwater environments[16]. These compounds are exclusively found in the heterocyte cell envelope of $N_2$-fixing heterocytous cyanobacteria[17–19], which are common components of the phytoplankton community in freshwater environments worldwide[20]. HGs consist of sugar moieties bound to even-numbered alkyl side chains with 26–32 carbon atoms containing either hydroxyl (diols, triols) or additional ketone groups (keto-ols, keto-diols; see Supplementary Fig. 1)[17]. The distribution of HGs has been shown previously to vary between different cyanobacterial orders and families[17,18], facilitating studies of community compositions of cyanobacteria in freshwater environments. In addition, the relative proportion of HGs systematically varies as a function of growth temperature, with $HG_{26}$ diols increasing and $HG_{26}$ keto-ols decreasing in abundance with increasing growth temperature in cultured cyanobacteria[21]. This pattern has been interpreted as a mechanism to constrain the diffusion of atmospheric gases into the heterocyte to protect the oxygen-sensitive enzyme nitrogenase and thus allow for biological $N_2$ fixation[22]. Very similar changes in the abundance of $HG_{26}$ diols and $HG_{26}$ keto-ols have been observed in surface waters of Lake Schreventeich (northern Germany). These changes were quantitatively expressed by the $HDI_{26}$ (heterocyte diol index of 26 carbon atoms), which tracked seasonal changes in SWT and evidenced that $HDI_{26}$ values in lake surface sediments capture a late summer temperature signal, during which maximum in-lake productivity was observed[16]. Significant differences in the composition of HGs have also been noted between polar microbial mats (dominated by HG keto-ols and keto-diols) and subtropical freshwater environments (dominated by HG diols and triols)[19]. Although there is compelling evidence that ambient temperature exerts a major control on the relative abundance of individual HGs, highlighting their potential in palaeoclimate investigations, no study has systematically examined HG distribution patterns in lakes of varying temperature regimes. This severely limits our understanding of the potential of HGs for studying past SWT in freshwater environments and reconstructing continental climate change.

Here, we investigate the spatial variability of HGs (in particular $HG_{26}$ diols and $HG_{26}$ keto-ols) in surface sediments of 46 tropical East African lakes in comparison to eight other globally distributed lakes and ponds to resolve the relationship between $HDI_{26}$ and environmental (SWT, pH, oxygen concentration), physical (lake depth, size, surface area) as well as biological parameters (productivity, community composition of cyanobacteria). In addition, the $HDI_{26}$ was for the first time applied to a sediment record from Lake Tanganyika (Tanzania), which allowed a comprehensive study of temperature change in tropical East Africa during the past ~37,000 years. Our data demonstrate the potential of HGs to reconstruct past lacustrine SWT and thereby act as a tool for the quantitative assessment of continental climate change.

## Results

**HG distributions in East African lake surface sediments.** We analyzed HG distribution patterns in surface sediments of 46 tropical East African lakes located on an altitudinal transect from 615 to 4504 m above sea level (masl; Fig. 1; Supplementary Table 1). The lakes show large physical, chemical, hydrological, and environmental gradients with maximum water depths differing between 0.3 and 1470 m, surface water pH varying from 3.8 to 9.8 and mean annual SWT ranging from 5.7 to 27.9 °C (Supplementary Table 1)[23,24]. Fourteen different HGs, ranging in chain length from 26 to 32 carbon atoms and each consisting of one to four structural isomers, were identified in the East African lake surface sediments (Fig. 2; Supplementary Table 2). Distribution patterns of HGs significantly varied along the altitudinal gradient. Hierarchical clustering of HGs and comparison with HG signatures found in cultured cyanobacteria[17–19,25] allowed the definition of three biozones, each characterized by distinct HG distributions and cyanobacterial communities as well as altitudinal ranges (Fig. 3). High relative abundances of $HG_{26}$ diols and $HG_{26}$ keto-ols, as well as $HG_{28}$ diols and $HG_{28}$ keto-ols, are specific for low-elevation lakes of Biozone 1 (615 to ~1550 masl). Both types of HGs have been reported previously from non-branching cyanobacteria belonging to the Nostocaceae, such as *Anabaena* spp. and *Anabaenopsis* spp.[18], which are common components of the phytoplankton community in equatorial African lakes at low altitudes[26,27]. Mid-elevation lakes of Biozone 2 (~1550 to ~4300 masl) are characterized by a higher diversity of HGs. Increased fractional abundances of $HG_{28}$ triols and $HG_{28}$ keto-diols as well as of $HG_{30}$ triols and $HG_{30}$ keto-diols suggest a higher contribution of mat-forming heterocytous cyanobacteria belonging to the Rivulariaceae (e.g. *Calothrix* spp.)[18,25] and Scytonemataceae (e.g. *Scytonema* spp.)[17,25]. $HG_{30}$ triols and $HG_{30}$ keto-diols increase in abundance in the high-elevation lakes, which are mostly shallow and oligotrophic tarns. In these freshwater environments of Biozone 3 (>4300 masl), $HG_{30}$ triols and $HG_{30}$ keto-diols constitute the majority of HGs and point to a predominant contribution of Scytonemataceae. The distribution of HGs thus indicates significant altitude-driven changes in the community composition of heterocytous cyanobacteria in the East African lakes.

$HG_{26}$ diols and $HG_{26}$ keto-ols, each consisting of two structural isomers, were present in all of the East African lake surface sediments (Fig. 3). The $HG_{26}$ diol isomers were most abundant in the low-elevation lakes with the most dominant isomer contributing on

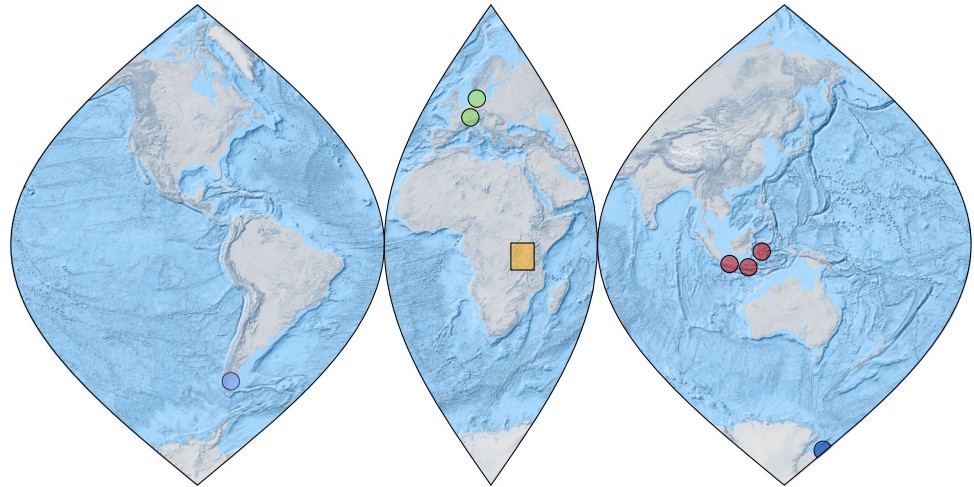

**Fig. 1 World map showing the locations of freshwater environments studied for their content of heterocyte glycolipids.** Orange square marks the area from which surface sediments of tropical East African lakes have been collected. This includes lakes from the Rwenzori Mountains (Uganda), Mt. Kenya (Republic of Kenya) and East African rift valley lakes that are located on an altitudinal transect from 615 to 4504 masl. Red dots = tropical Lake Towuti (Indonesia), Lake Klakah (Indonesia), and Lake Lading (Indonesia). Green dots = temperate Lake Constance (Germany) and Lake Schreventeich (Germany)[16]. Light blue dot = subpolar Laguna Potrok Aike (Argentina). Dark blue dot = Antarctic meltwater ponds.

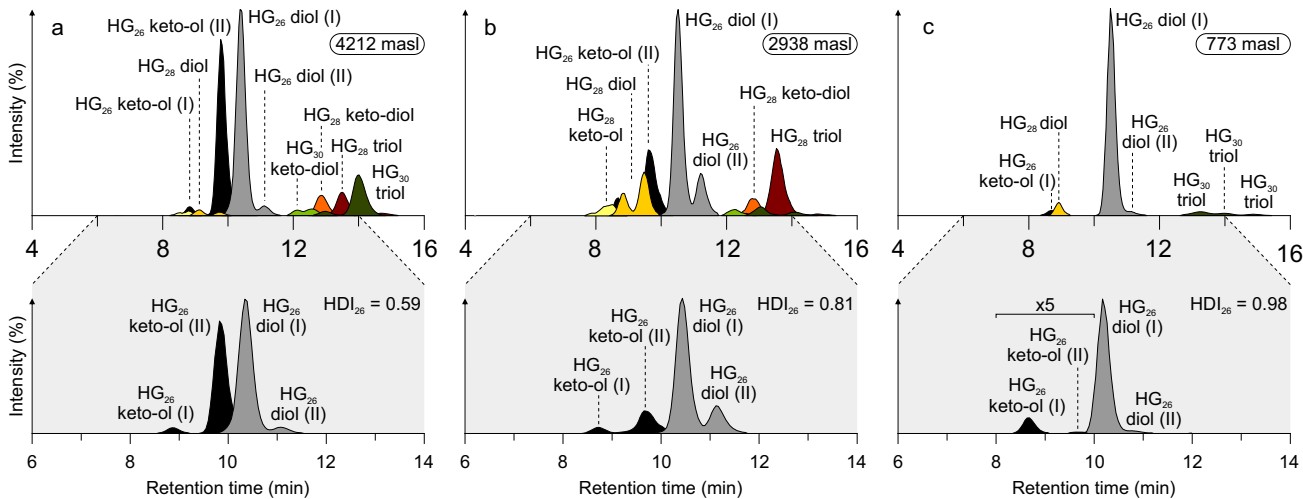

**Fig. 2 Composite mass chromatograms showing the distribution of heterocyte glycolipids (HGs) in East African lake surface sediments.** Representative samples include **a** high-elevation Lake Hohnell (4212 masl; 7.9 °C), **b** mid-elevation Lake Bandasa (2938 masl; 17.2 °C), and **c** low-elevation Lake Tanganyika (773 masl; 25.7 °C). Insets show the distribution of $HG_{26}$ diols and $HG_{26}$ keto-ols in the respective samples. Note the increase in the relative abundance of $HG_{26}$ diols compared to $HG_{26}$ keto-ols with decreasing altitude and increasing lake surface water temperature as well as the concomitant increase in $HDI_{26}$ values.

average $0.85 \pm 0.04$ to the summed fractional abundance of $HG_{26}$ diols and $HG_{26}$ keto-ols. However, its abundance declined gradually along the altitudinal gradient and in lakes >4000 masl, the fractional abundance of the most dominant $HG_{26}$ diol was on average $0.57 \pm 0.01$ (Fig. 4; Supplementary Table 3). By contrast, the most abundant isomer of the $HG_{26}$ keto-ols contributed with $0.04 \pm 0.01$ only minor to the summed fractional abundance of $HG_{26}$ diols and $HG_{26}$ keto-ols in the low-elevation lakes. It increased, however, gradually in abundance with altitude to a maximum of $0.33 \pm 0.05$ of the summed fractional abundance of $HG_{26}$ diols and $HG_{26}$ keto-ols in the high-elevation lakes (Fig. 4). The $HDI_{26}$ (heterocyte diol index of 26 carbon atoms = $HG_{26}$ diol/[$HG_{26}$ diol + $HG_{26}$ keto-ol]), a means to quantitatively express changes in the relative abundances of $HG_{26}$ diols compared to $HG_{26}$ keto-ols[16], varied from 0.59 in high-elevation Lake Hohnell (4212 masl) to 0.99 in low-elevation Lake Albert (615 masl; Supplementary Table 3).

**HG distributions in tropical to polar lakes and ponds**. In order to assess whether HGs are found globally and whether their distribution shows significant variation between climate zones, we investigated a set of surface sediments collected from eight polar to tropical freshwater systems (Fig. 1). These lakes and ponds were selected based on the availability of direct water temperature measurements, which allowed investigating the effect of seasonality on HG distribution patterns and consequently the reconstruction of SWT. $HG_{26}$ diols were abundant in the three tropical Indonesian lakes (Klakah, Lading, and Towuti). The most dominant $HG_{26}$ diol isomer on average amounted to $0.89 \pm 0.05$ ($0.01 \pm 0.01$ for the $HG_{26}$ keto-ol) of the summed fractional abundances of $HG_{26}$ diols and $HG_{26}$ keto-ols (Supplementary Table 3). In temperate Lake Constance (southern Germany), its fractional abundance was 0.76 (0.16 for the $HG_{26}$ keto-ol). A similar fractional abundance of 0.79 for the $HG_{26}$ diol (0.21 for

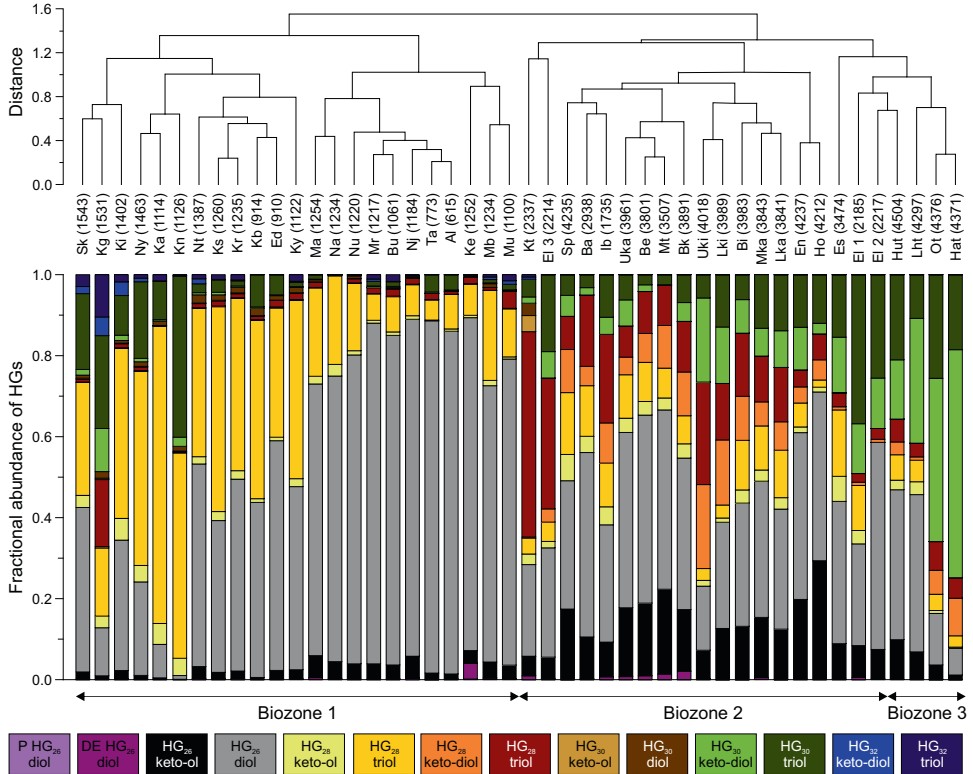

**Fig. 3 Hierarchical clustering of East African lakes based on the sedimentary distribution of heterocyte glycolipids (HGs).** The distance in the similarity of HG patterns indicates that three biozones, each characterized by distinct cyanobacterial communities, exist along the altitudinal gradient. Biozone 1 comprises low-elevation lakes (615 to ~1550 masl) and shows abundant $HG_{26}$ diols and $HG_{28}$ diols that are specific to nostocalean cyanobacteria belonging to the genera *Anabaena* and *Anabaenopsis*[17,18]. Biozone 2 constitutes of mid-elevation lakes (~1550 masl to ~4300 masl). Surface sediments of these lakes contain more diverse HG distribution patterns and higher relative abundances of $HG_{28}$ triols and $HG_{30}$ triols. The latter are found in heterocytous cyanobacteria of the benthic mat-forming families Rivulariaceae (e.g. *Calothrix* spp.)[18,19] and Scytonemataceae (e.g. *Scytonema* spp.)[25]. Biozone 3 (>4300 masl) includes shallow, high-elevation tarns that are characterized by abundant $HG_{30}$ triols and $HG_{30}$ keto-diols, which points to a predominant contribution of mat-forming heterocytous cyanobacteria of the genus *Scytonema*[25]. Note that only the summed fractional abundance of individual HGs is shown as the overall HG distribution pattern is specific for the cyanobacterial community composition. Lake abbreviations are provided in Supplementary Table 1. Numbers in brackets indicate the altitudinal position of the lake systems.

the $HG_{26}$ keto-ol) has been reported previously in temperate Lake Schreventeich (northern Germany)[16]. The fractional abundance of the most dominant $HG_{26}$ diol was 0.52 (0.27 for the $HG_{26}$ keto-ol) in surface sediments of subpolar Laguna Potrok Aike, which is in a similar order of magnitude as observed in the East African high-elevation lakes with SWTs of 5–7 °C. $HG_{26}$ diols and $HG_{26}$ keto-ols were also abundant in polar meltwater ponds of western Antarctica with fractional abundances of the most dominant $HG_{26}$ diol ranging from 0.27 (0.25 for the $HG_{26}$ keto-ol) in Orange Pond to 0.29 (0.24 for the $HG_{26}$ keto-ol) in Conophyton Pond.

Similar to the low-altitude East African lakes, $HDI_{26}$ values ranged from 0.99 to 1.00 in the warm Indonesian lakes, were intermediate with 0.78 (Lake Constance) to 0.79 (Lake Schreventeich)[16] in the two temperate European lakes, but were substantially lower with 0.66 in subpolar Laguna Potrok Aike (Supplementary Table 3). In the Antarctic meltwater ponds with water temperatures close to the freezing point, $HDI_{26}$ values ranged from 0.51 to 0.55.

**HG distributions in tropical Lake Tanganyika.** Surface sediments of Lake Tanganyika contained high fractional abundances of $HG_{26}$ diols (0.87) and $HG_{26}$ keto-ols (0.02), which together comprised the majority of all HGs (Fig. 2; Supplementary Tables 2 and 4). $HG_{28}$ diols (0.05) and $HG_{28}$ keto-ols (<0.01) occurred only in minor relative abundances. Traces of $HG_{28}$ triols

(0.02) and $HG_{28}$ keto-diols (<0.01), as well as $HG_{30}$ triols (0.04) and $HG_{30}$ keto-diols (<0.01) were also detected. The most dominant $HG_{26}$ diol isomer amounted to 0.90 ± 0.01 of the summed fractional abundance of $HG_{26}$ diols and $HG_{26}$ keto-ols, while the most dominant isomer of the $HG_{26}$ keto-ols contributed no more than 0.01 ± 0.01 (Supplementary Table 3). A $HDI_{26}$ value of 0.98 was determined for the lake surface sediment.

Subsurface sediments of Lake Tanganyika, representing the last ~37,000 years of East African climate history, were collected from sediment core NP04-KH04-4A-1K[28]. Fractional abundances of the most dominant $HG_{26}$ diol ranged from 0.83 during the last glacial maximum (LGM) to 0.90 at present, while the fractional abundance of the predominant $HG_{26}$ keto-ol varied concomitantly from 0.08 to 0.01, respectively (Supplementary Table 5). $HDI_{26}$ values were as low as 0.91 during the LGM and increased to 0.98 in modern Lake Tanganyika (Supplementary Table 5).

**Discussion**

Our biomarker data demonstrates that HGs are present in all of the studied freshwater environments. This is consistent with their presence in North American[29] and European freshwater lakes[16,19] as well as microbial biofilms collected from Antarctica[19], Iceland[30], and Svalbard[31]. These studies have shown that $HG_{26}$ diols and $HG_{26}$ keto-ols are most widespread[16–18,29], in agreement with the ubiquitous presence of cyanobacteria belonging to the Nostocaceae (such as *Anabaena* spp.,

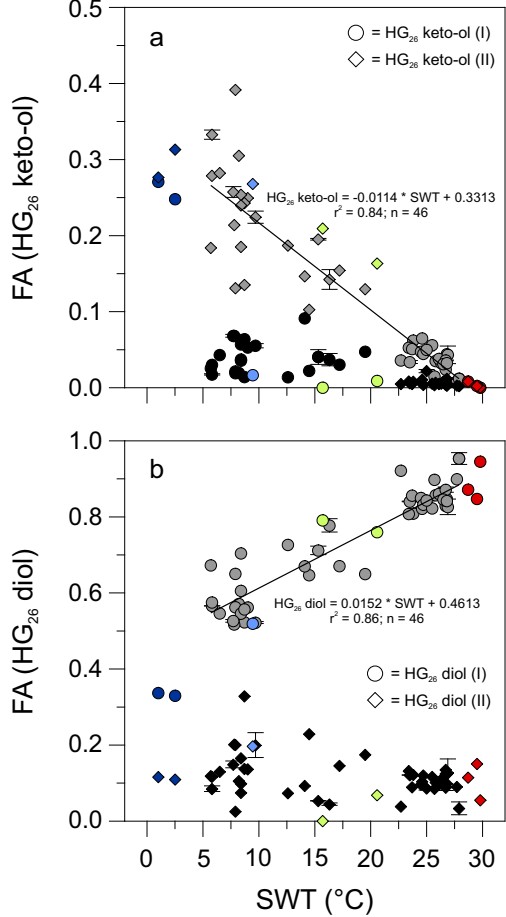

**Fig. 4 Fractional abundances of heterocyte glycolipids in East African lakes as well as other polar to tropical lakes and ponds.** Regression analysis demonstrates that fractional abundances (FA) of (**a**) $HG_{26}$ keto-ols are negatively and that of (**b**) $HG_{26}$ diols are positively correlated with surface water temperature (SWT). The most dominant structural isomers of both component classes in the East African (indicated in gray) and polar to tropical lake surface sediments showed the strongest correlation with SWT and were used for the calculation of the $HDI_{26}$. Fractional abundances of $HG_{26}$ keto-ols and $HG_{26}$ diols found in surface sediments of polar to tropical lakes and ponds are shown for comparison and express similar compositional changes across the investigated temperature interval as observed for the East African lake sample set. Red circles and diamonds = tropical Indonesian lakes (Lake Klakah, Lake Lading, and Lake Towuti); Green circles and diamonds = temperate European lakes (Lake Constance and Lake Schreventeich[16]; Light blue circle and diamond = subpolar Laguna Potrok Aike; Dark blue circles and diamonds = Antarctic meltwater ponds. Error bars indicate the SD of the fractional abundances based on replicate sample analyses.

*Aphanizomenon* spp., *Nodularia* spp.) as part of the phytoplankton community in lakes worldwide[20]. Both HGs were also most widespread and abundant in the East African ($0.53 \pm 0.25$ of all HGs) as well as tropical to polar lake surface sediments and ponds ($0.37 \pm 0.35$ of all HGs; Supplementary Table 2). Similar to previous culture studies[18,21] and observations from a time series experiment[16], the fractional abundances of $HG_{26}$ diols and $HG_{26}$ keto-ols showed significant changes along the altitudinal gradient of East African lakes and between lakes of different climate zones (Figs. 2–4; Supplementary Tables 2 and 3). In general, $HG_{26}$ diols increased in abundance with decreasing altitude and increasing SWT as well as from polar to tropical latitudes, while $HG_{26}$ keto-ols showed an opposing trend.

Bivariate correlation analysis demonstrates that changes in the fractional abundance of the most dominant $HG_{26}$ diol in the East African lakes are positively correlated with SWT ($r = 0.953$; $p < 0.0001$; $n = 42$; Supplementary Table 6). In contrast, variations in the fractional abundance of the most dominant $HG_{26}$ keto-ols show a strong negative and statistically significant correlation with SWT ($r = -0.979$; $p < 0.0001$; $n = 42$). Similar correlations between the fractional abundance of $HG_{26}$ diols and $HG_{26}$ keto-ols and temperature have been observed previously in cultured cyanobacteria[21] and environmental samples[16,19]. Statistical analysis also indicates a strong positive correlation of changes in $HDI_{26}$ values with SWT ($r = 0.975$; $p < 0.0001$; $n = 42$). In addition, the $HDI_{26}$ shows significant correlations with elevation, mean annual air temperature (MAAT) and bottom water temperature (BWT; Supplementary Table 6). Significant but generally weak correlations are observed with water depth, conductivity, surface (SW pH) as well as bottom water pH (BW pH) and bottom water dissolved oxygen concentrations (BW DO; Supplementary Table 6). These parameters, however, are also all significantly correlated with SWT, suggesting a minor or only indirect influence of parameters other than temperature on the abundance of both HGs and the calculation of the $HDI_{26}$. Partial correlation analysis indeed demonstrates that none of these variables (except elevation and MAAT) show a significant correlation with the abundances of $HG_{26}$ diols and $HG_{26}$ keto-ols after the effect of SWT as a control variable is removed (Supplementary Table 7). Generally low or absent correlations of HG abundances with environmental parameters, such as pH and DO, have been reported previously from lake systems[16,29], providing additional evidence that SWT, either by controlling the amount of oxygen dissolved in lake water or the rate of oxygen diffusion into the heterocyte, regulates the synthesis of HGs. These structural changes in the heterocyte cell envelope likely alter the properties of the gas diffusion barrier against the entry of atmospheric gases (including $O_2$) into the heterocyte in order to allow for optimal $N_2$ fixation[21]. No correlations with lake surface area, water column conductivity, SW DO or organic matter content, and the abundance of $HG_{26}$ diols and $HG_{26}$ keto-ols or the $HDI_{26}$ are observed (Supplementary Table 6).

The $HDI_{26}$ employs the relative abundances of $HG_{26}$ diols and $HG_{26}$ keto-ols to infer changes in water temperature[16]. In the East African as well as globally distributed lakes and ponds, both components occur as two structural isomers (Supplementary Table 3). Their retention time is indistinguishable from that of $HG_{26}$ diols and $HG_{26}$ keto-ols in cultured cyanobacteria, arguing for a direct biological origin of these components and against sedimentary production via rearrangement reactions. This may suggest that there is little impact of diagenetic overprinting on the distribution pattern of $HG_{26}$ diols and $HG_{26}$ keto-ols in sediments. For the two structural modifications of the $HG_{26}$ diol, the strongest correlation with SWT was noted for the most dominant early eluting isomer ($r = 0.953$; $p < 0.0001$; $n = 42$; Fig. 2). In the case of the two varieties of the $HG_{26}$ keto-ol, a generally strong correlation with SWT was observed with the later eluting isomer ($r = -0.964$; $p < 0.0001$; $n = 42$), which was dominating in lakes of Biozones 2 and 3. In all lakes of Biozone 1, however, the early eluting isomer of the $HG_{26}$ keto-ol was predominant (Fig. 4). Changes in the relative abundance of individual HG isomers in response to variations in growth temperature were reported previously from the thermophilic cyanobacterium *Mastigocladus* sp. and considered as an alternative means for fine-tuning the properties of the gas diffusion barrier by synthesizing HGs with different spatial dimensions of the sugar moiety[30]. This finding is corroborated by the changes in the dominance of the $HG_{26}$ keto-ol isomers observed here. Taking these observations into account, the best correlation with SWT ($r = -0.979$; $p < 0.0001$; $n = 42$)

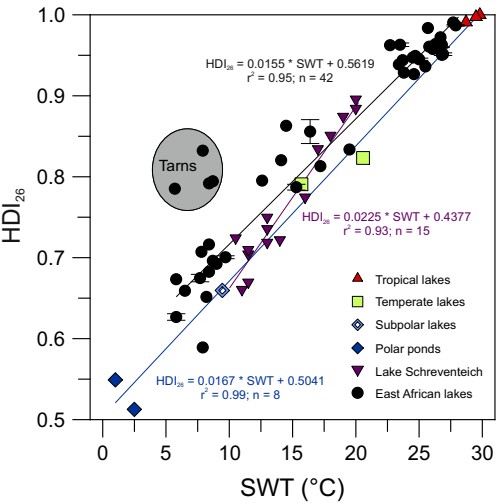

**Fig. 5 Cross plot of HDI$_{26}$ (heterocyte diol index of 26 carbon atoms) values versus surface water temperature (SWT) in East African and other globally distributed lakes and ponds.** Note that the four high-elevation tarns of Biozone 3 (>4300 masl) are not included in the calculation of the correlation coefficient of East African lakes (black regression line and calibration). For comparison, regression lines of SWT and HDI$_{26}$ values extracted from surface sediments of tropical to polar lakes and ponds (blue regression line and calibration) as well as water column samples (purple regression line and calibration) of temperate Lake Schreventeich[16] are displayed, showing regional differences in the correlation between the synthesis of heterocyte glycolipids and water temperatures. Black dots = African lakes; red triangles = tropical Indonesian lakes; green squares = temperate European lakes; open blue diamond = subpolar Laguna Potrok Aike; dark blue diamonds = Antarctic meltwater ponds; purple triangles = water column samples of Lake Schreventeich[16]. Error bars indicate the SD based on replicate sample analyses.

was obtained by selecting the most abundant HG$_{26}$ keto-ol isomer in each sample for the calculation of the HDI$_{26}$. Our data thus provides evidence for a significant correlation between SWT and the relative abundances of HG$_{26}$ diols and HG$_{26}$ keto-ols in the East African lake surface sediments and, together with previous culture experiments[18,21], indicate that the HDI$_{26}$ is primarily controlled by temperature-induced changes in the composition of the heterocyte cell envelope.

Regression analysis indicates that the HDI$_{26}$ and SWT are best correlated using the following linear equation in the East African lake surface sediments: HDI$_{26}$ = 0.0155 × SWT + 0.5619 ($r^2$ = 0.95, $n$ = 0.42; RSME = 1.8 °C; Fig. 5). Although a strong relationship between the HDI$_{26}$ and SWT is evident across the dataset, some of the lakes clearly deviate from the general trend, including Eldoret Nakuru 2 (2217 masl) and Nyamswiga (1463 masl). Some of the scatter may be related to shifts in the cyanobacterial community between different biozones or be caused by multiple cyanobacterial species with different blooming periods and/or different temperature responses towards the synthesis of HGs within a lake. These effects, however, may be partially counteracted by the time-integrating nature of sediment archives. As the sedimentary HG signal represents a multi-year average, this may reduce the overall impact of interannual variability and species differences on local to regional scales and contribute to the generally strong correlation of the HDI$_{26}$ with SWT observed in the East African lakes.

To investigate whether a similar correlation exists on a global scale, we also applied the HDI$_{26}$ to surface sediments of lakes and ponds from polar to tropical climate zones. This approach resulted in the following strong linear global correlation: HDI$_{26}$ = 0.0167 ×

SWT + 0.5041 ($r^2$ = 0.99, $n$ = 8; RSME = 1.7 °C; Fig. 5). Again, the most dominant HG$_{26}$ diol and HG$_{26}$ keto-ol isomers were used for the calculation of the HDI$_{26}$ as this iteration of the proxy yielded the strongest correlation with SWT. Analysis of covariance demonstrates that the East African lake and the calibration of the globally distributed lakes have no statistically significant difference in slope ($p$ = 0.17) but do differ in the intercepts of the regression lines ($p$ = 0.007). Very similar offsets in HDI$_{26}$ values have been observed in cultured cyanobacteria[21], suggesting that on a global scale species-specific effects in geographical distinct regions may become more pronounced and changes in the cyanobacterial community may require regional calibrations for the accurate reconstruction of SWTs using the HDI$_{26}$. This is also suggested by the difference in slope and intercept of the transfer function established for water column samples from Lake Schreventeich[16] and those reported here (Fig. 5). Together these observations argue for additional environmental and cultures studies to further constrain the differential control of temperature on the HDI$_{26}$.

Replicate analysis of selected lake surface sediments across the entire sample set revealed that the analytical accuracy with which the HDI$_{26}$ can be determined is on average ± 0.002. This equals to an analytical error of ±0.2 °C. A larger uncertainty in the determination of past lake water temperatures is usually embedded in the calibration function. The root-mean-square error (RSME), a measure for the uncertainty in temperature prediction, of the East African and global lake transfer functions is 1.8 and 1.7 °C, respectively (Supplementary Table 3). Moreover, the RSME does not evidence any apparent deviation from linearity across the investigated temperature interval. This suggests that the proxy is not affected by differential lipid synthesis towards the extreme ends of the temperature spectrum (Supplementary Table 3). The accuracy with which SWTs can be reconstructed from lacustrine archives using the HDI$_{26}$ is thus in a similar order of magnitude compared to other commonly applied geochemical temperature proxies[32,33] and bioindicators[34].

Seasonality and habitat depth strongly control biological productivity and have been demonstrated to impact proxy-based climate reconstructions[35,36]. Due to the close equatorial position of the East African lakes, variation in SWT over a full annual cycle is only minor and ranges between 2 and 3 °C[37,38]. In contrast, temperate to polar lakes are characterized by a significantly larger seasonality and more pronounced changes in temperature and productivity. In temperate Lake Constance (southern Germany), measured SWTs range from 5.1 to 25.9 °C (annual mean of 12.3 °C). Surface sediments of this lake yielded a HDI$_{26}$-based SWT of 19.1 °C (Supplementary Table 3). This ~7 °C bias towards higher water temperatures compared to the annual mean implies that the HDI$_{26}$–reconstructed SWT records the summer maximum of cyanobacterial activity observed in Lake Constance. A very similar bias has been reported previously from temperate Lake Schreventeich (northern Germany), in which the HDI$_{26}$ recorded a late summer temperature signal[16]. Such an observation is also in agreement with a growth optimum of cyanobacteria shifted to higher temperatures compared to most eukaryotic algae[39] and reduced availability of combined nitrogen in surface waters during summer, which promotes growth of diazotrophic cyanobacteria[40]. This suggests that sedimentary HG distribution patterns in temperate to polar lakes and ponds are biased by increased summer productivity and that the HDI$_{26}$ in such settings does reflect a seasonal and not a mean annual temperature signal. In contrast, changes in habitat depths of heterocytous cyanobacteria are likely to exert only a minor control on the HDI$_{26}$ signal. Heterocytous cyanobacteria can control their buoyancy and to some extent actively regulate their position in the water column, where they commonly form a dense cover on or close to the surface[41]. Extensive shading of the underlying

water column during such bloom events can preclude major vertical migration of heterocytous cyanobacteria and restricts their presence close to the surface. In contrast to other organic temperature proxies (such as the $TEX_{86}$), the habitat depth of the biological sources of HGs is thus comparatively well-constrained and limited to the uppermost body of the epilimnion.

Despite the overall good correlation between SWT and the $HDI_{26}$, four of the East African lakes expressed relatively high $HDI_{26}$ values compared to measured water temperatures (Fig. 4). These all comprise the clear and mostly shallow high-elevation tarns of Biozone 3 (>4300 masl). HG profiles of these lakes are distinctly different from those of other East African lakes. Based on comparison with HG distribution patterns derived from culture experiments[17,25], benthic cyanobacteria of the genus *Scytonema* are likely most dominant in these tarns, which points to a significant shift in the cyanobacterial community. In the absence of large seasonal temperature variation, the observed bias in calculated $HDI_{26}$ values may either be explained by a species-specific effect with a different response in lipid synthesis to temperature and/or induced by heating of cell surfaces due to light energy absorption via photosynthetic and photoprotective pigments. In situ measurements and remote sensing demonstrated that through this mechanisms, cyanobacterial surface blooms can increase temperature locally by 1–5 °C above ambient water[42–44]. Local heating of cell surfaces of pelagic cyanobacteria or shallow benthic mats may, at least in parts, explain the unexpectedly high $HDI_{26}$ values observed in the high-elevation tarns. Other factors, including increased UV radiation or contributions of terrestrial heterocytous cyanobacteria with potentially different HG responses to temperature, may also affect HG distributions. Although such contributions—in particular during bloom events —may only be little, this needs to be explored in future studies.

In order to evaluate the potential of the $HDI_{26}$ in reconstructing past variations in continental climates, we investigated HG distribution patterns in sediment core NP04-KH04-4A-1K collected from tropical Lake Tanganyika. The core was obtained from the distal margin of the Kayla Platform, located in the central part of the lake, in a water depth of ~330 m. The recovered sediment sequence consists of a succession of alternating diatomaceous oozes and massive to silty clays[28]. $^{14}$C AMS radiocarbon dating and stratigraphic correlation with parallel core KH03 indicate that the 729 cm-long sediment sequence covers the last ~37,000 years of East African climate history[45]. Lake Tanganyika surface sediments contained abundant $HG_{26}$ diols and $HG_{26}$ keto-ols. The $HDI_{26}$-reconstructed SWT of the surface sediment using our East African lake calibration function is 27.2 °C. This temperature is well within the measured annual SWT variation of Lake Tanganyika, which ranges from 24.1 to 28.5 °C based on a 5-year average for the time period from 2004 to 2009[38]. The sedimentary HG distribution thus seems to record the temperature of highest cyanobacterial productivity but also in this tropical lake seems to be biased towards a warmer water temperature signal, despite the comparatively low annual SWT variability.

In the Lake Tanganyika sediment sequence, HG distribution patterns are highly variable and shift from a dominance of $HG_{30}$ triols (0.73 ± 0.19) and $HG_{30}$ keto-diols (0.05 ± 0.04) during the LGM to the predominance of $HG_{26}$ diols (0.87 ± 0.01) and $HG_{26}$ keto-ols (0.02 ± 0.01) observed in the lake surface sediment (Supplementary Table 4). These variations in the distribution of HGs are likely related to changes in the cyanobacterial community and during the LGM may arise from the spread of mat-forming cyanobacteria in the littoral and subsequent sediment transport to the profundal in a lake with a water level 250–300 m lower than present[28,46].

$HDI_{26}$-reconstructed SWTs of Lake Tanganyika show significant variations over time that relate to global climate trends as well as abrupt climate change events (Fig. 6; Supplementary Table 5). At the base of the record (~37,000 years BP), the $HDI_{26}$-calculated SWT was 23.8 °C. Over the following 10,000 years, the lake water cooled by 1.3 °C. During the LGM, the $HDI_{26}$-inferred temperature averaged ~22.5 °C and started to increase between 19,000 and 15,000 years BP, in line with rising global atmospheric $CO_2$ concentrations (Fig. 6)[47]. SWT continued to increase during the deglacial period and the early Holocene to yield a maximum of 26.5 °C during the mid-Holocene (~4200 years BP), during a $CO_2$ minimum but in keeping with other records from the African tropics[48]. Thereafter, $HDI_{26}$ temperatures decreased gradually by ~1.8 °C and started to increase again after ~1,700 years BP to reach a temperature maximum of 27.2 °C in modern Lake Tanganyika.

Our $HDI_{26}$-SWT record indicates an overall ~4.1 °C warming in tropical East Africa from the LGM until the onset of the industrial period. This warming is generally similar to other late Pleistocene and Holocene climate records obtained from East African Rift Valley lakes, such as Lake Malawi[49,50] and Lake Turkana[51]. Moreover, our temperature record is within the range of estimated deglacial warming in low-elevation tropical East Africa of 4 ± 2 °C based on pollen[52] and lipid palaeothermometers[53]. Besides the superimposed long-term temperature trend, the $HDI_{26}$ also provides evidence for abrupt climate change events. Though the resolution of our record is too low to study millennial-scale changes in detail, the $HDI_{26}$ indicates a decline in SWT by ~0.4 °C about ~12,800 years BP, roughly coincident with the onset of the Younger Dryas (YD) cold event[54] (Fig. 6). Equivalent cooling during the YD chronozone has also been documented at other locations in East Africa, such as Lake Albert[55], Lake Rutundu[56], and Lake Malawi[49], but they are controversial as they are absent in other temperature records from this region. Our $HDI_{26}$-based temperature record, however, suggests that connections between low-latitude and high-latitude may indeed impact the climate of tropical East Africa.

Although our $HDI_{26}$-SWT record is very similar in terms of trends and magnitude compared to the ~4 °C inferred warming from the late Pleistocene to late Holocene based on the $TEX_{86}$ lipid palaeothermometer in Lake Tanganyika[57], it deviates during the youngest part of the sediment sequence. Our record indicates warming during the early Holocene that culminated in a maximum water temperature of 26.5 °C in Lake Tanganyika during the mid-Holocene. The $TEX_{86}$ indicates significantly lower water temperatures during this time period[45] with a temperature offset of ~1.6 °C between both proxies. Previous comparison of the $TEX_{86}$ signal in suspended particulate matter with measured water column temperatures indicates that the $TEX_{86}$ underestimates SWTs by ~2 °C in modern Lake Tanganyika[38]. This offset is in a similar order of magnitude as observed between the mid to late-Holocene $TEX_{86}$-based and $HDI_{26}$-reconstructed water temperatures and may causally be related to a primary residence depth of GDGT-producing Thaumarchaeota not at the surface but the oxycline that is currently situated at a water depth of ~100 m in Lake Tanganyika[58]. Changes in the lake's hydrology and increased lake mixing during a cooler and drier glacial may be responsible for the close progression of both proxies by reducing the temperature difference between the residence depths of heterocytous cyanobacteria thriving at the surface of Lake Tanganyika and deep-dwelling Thaumarchaeota. In contrast, increased lake stratification and lesser mixing of the water column has been inferred for the lake during the Holocene[59] and together with changes in oxycline depth[46] and/or seasonal productivity is likely to explain the here observed offset between both proxy records. Nevertheless, the overall similarity of both records, as well as the good agreement of the timing and magnitude of deglacial warming compared to other low-elevation climate records from this region, including those from Lake Malawi[49] and

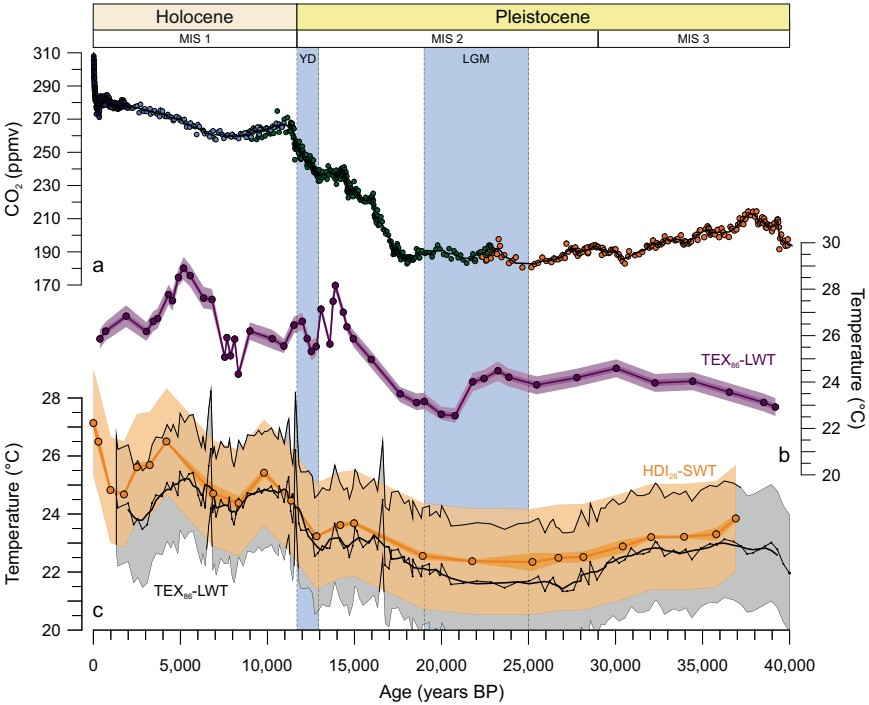

**Fig. 6 Downcore plot of HDI$_{26}$-reconstructed surface water temperatures in Lake Tanganyika in comparison with global $CO_2$ records and regional records of East African deglacial warming.** Profiles are plotted as follows: **a** Deglacial $CO_2$ record from the Dome Concordia (Dome C) and other ice cores[47]. **b** Tropical Lake Malawi TEX$_{86}$ lake water temperature (LWT) record[49,50]. Light and dark purple shaded areas denote 68% and 95% confidence intervals of the reconstruction based on bootstrapping of the respective calibration[56]. **c** HDI$_{26}$-based surface water temperature (SWT) record and TEX$_{86}$-LWT record[57] in tropical Lake Tanganyika. Light blue shading denotes the age range of the regional last glacial maximum (LGM), defined by $^{10}$Be ages of moraines from the Rwenzori Mountains[69], and the Younger Dryas (YD) cold event[54]. The gray shaded area indicates the calibration uncertainty associated with the TEX$_{86}$. The calibration uncertainty of the HDI$_{26}$ is indicated in light orange, while the analytical error based on replicate measurements is shown in dark orange.

Lake Victoria[60] as well as the Congo Basin[61], emphasizes the robustness of the HDI$_{26}$ in tracking continental climate change on regional scales.

Our data demonstrates that the HDI$_{26}$ is significantly correlated with modern SWT in East African lakes as well as other polar to tropical freshwater environments, indicating that the HDI$_{26}$ can be applied on a global scale and over a temperature range from 1 to ~28 °C. Its first application to a ~37,000-years sediment record from tropical Lake Tanganyika also shows that the HDI$_{26}$ constitutes an independent tool to access information on past SWTs and thus on continental climate change archived in fossil lacustrine sequences. The recent discovery of HGs in late Cretaceous sediments (80–90 Myr)[62] and sediments covering the Cretaceous–Palaeogene boundary (~66 Myr)[63] thus suggests that HG-based indices (such as the HDI$_{26}$) may allow reconstructing SWTs and continental climates throughout the Cenozoic and beyond.

## Methods

**Lakes and sediment sampling.** Surface sediments (in most cases 0–1 cm) were collected from 46 East African lakes located in the Rwenzori Mountains (Uganda, Democratic Republic of Congo) and Mt. Kenya (Kenya) but East African Rift Valley lakes were also included (see Supplementary Table 1). The lakes were situated on an altitudinal transect from 615 to 4504 masl and were selected to cover a wide range of physical, chemical, and hydrological parameters including, e.g. water temperature, lake size and depth, pH and productivity (see Supplementary Table 1 for a comprehensive description of environmental variables)[23,24]. Surface sediments from tropical Lake Towuti (Indonesia), Lake Lading (Indonesia) and Lake Klakah (Indonesia) were obtained in June 2013 using an Uwitec gravity corer. Surface sediments from temperate Lake Constance (Germany) were taken in summer 2016 also using an Uwitec gravity corer. Surface sediments of subpolar Laguna Potrok Aike (southern Patagonia, Argentina) were collected from short gravity core PTA02/4 taken in 2003 and stored at +4 °C thereafter. Microbial mats were collected from Conophyton Pond and Orange Pond on the McMurdo Ice Shelf (Antarctica) in January 2018 and stored at −20 °C thereafter. All samples were shipped to Christian-Albrechts-University on dry ice where they were stored

frozen until they were processed and analyzed for HG distribution patterns and HDI$_{26}$ values. The here generated data was complemented by previously published HG signatures and HDI$_{26}$ values from surface sediments and water column experiments in temperate Lake Schreventeich (Germany)[16].

Subsurface sediments from Lake Tanganyika were collected from piston core NP04-KH04-4A-1K taken on the Kalya Horst (S 6°42′, E 29°50′) at a water depth of ~330 m[28]. To minimize degradation of the sedimentary organic matter, the core was stored cold (+4 °C) at Brown University since its recovery in 2004. Solvent-cleaned spatulas were used to collect 5 mm-thick sediment slices, which were transferred to Whirl-Pak sample bags for transport, at an average resolution of ~1500 years. Together with the surface sediments, these samples were shipped to Christian-Albrechts-University on dry ice. Upon arrival, all sediments were lyophilized, ground to a fine powder using a solvent-cleaned agate pestle and mortar and stored frozen until further processing.

**SWTs of polar to tropical lakes.** All SWTs reported here were collected within the top 10 cm of the water column. SWT of East African lakes were taken during several field campaigns from 1996 to 2010 and varied from 5.7 °C in Hut Tarn (4504 masl) to 27.9 °C in Lake Albert (615 masl)[23,24]. Water temperatures of tropical Lake Towuti (Indonesia) were recorded from 2012 to 2014 with Hobo Tidbit data loggers (CiK Solutions, Germany) at 15-m depth intervals[64]. Over the time course of a year, SWT ranged from 28.9 °C in August 2013 to 30.4 °C in March 2013 with a mean annual temperature signal of 29.8 ± 0.5 °C. SWT of Lake Klakah (Indonesia) and Lake Lading (Indonesia) were determined during field work conducted in June 2013 and January 2014; these averaged 29.5 and 28.7 °C, respectively. Water temperatures of temperate Lake Constance (southern Germany) are continuously monitored by the environmental agency of the state Baden-Württemberg. From 2011 to 2015, SWT varied from 5.1 ± 0.3 °C in winter to 20.6 ± 1.0 °C in summer. The average annual SWT was 12.3 ± 0.7 °C. Water temperature measurements were conducted in Lake Schreventeich (northern Germany) from July to October 2014 using an Oxi 1970i instrument coupled to a CellOx325 oxygen probe (WTW, Germany)[16]. SWT of the small meromictic lake varied from 24.0 °C in early August to 10.5 °C in late October. Highest abundances of HGs, in conjunction with highest in-lake productivity, were observed in the first half of September with SWT averaging 15.7 °C. Temperatures of subpolar Laguna Potrok Aike were recorded at seven different water depths from 2014 to 2017 using M-08TR Minilog thermistors (Vemco Ltd., Canada) attached to a mooring[65]. The lake SWT over the 4-year period varied from 3.7 ± 0.2 °C in September to 10.7 ± 0.7 °C in February. The mean SWT signal was 7.1 ± 0.4 °C and the summer SWT

averaged $9.5 \pm 0.5\,°C$. Water temperatures of the Antarctic meltwater ponds were determined during the field sampling campaign in January 2018 and ranged between $1.0\,°C$ for Conophyton Pond and $2.5\,°C$ for Orange Pond. Water temperatures were collected in austral summer and reflect the time of highest productivity in the ponds, while temperatures were below freezing point for the remainder of the year[66], preventing activity of cyanobacteria and the synthesis of HGs.

**Heterocyte glycolipid extraction and analysis**. Between 0.5 and 2 g of dried and homogenized lake surface and subsurface sediments were extracted using a modified Bligh and Dyer procedure[16]. For this, a known volume of a single-phase solvent mixture of methanol (MeOH)/dichloromethane (DCM)/phosphate buffer (2:1:0.8; v:v:v) was added to the sediments, which were then extracted in an ultrasonic bath for 10 min. After centrifugation ($4000 \times g$, 10 min), the supernatant was collected and the residue extracted three more times with the above described solvent mixture. DCM and phosphate buffer were added to the combined extracts to a volume ratio of 1:1:0.9 (v:v:v), resulting in phase separation. The bottom layer, containing the organic fraction, was collected after centrifugation ($1000 \times g$, 5 min) and the remaining MeOH/phosphate buffer phase was washed twice with DCM. The bulk of organic solvents was removed by rotary evaporation and the Bligh and Dyer extract (BDE) dried under a gentle stream of nitrogen. Aliquots of the BDEs were re-dissolved in a solvent mixture of $n$-hexane:2-propanol:$H_2O$ (71:27:1, v:v:v) to a concentration between 3 and $9\,mg\,mL^{-1}$ dependent on the abundance of the target analyte, filtered through a $0.4\,\mu m$ mesh-size regenerated cellulose syringe filter and subjected to high-performance liquid chromatography coupled to electrospray ionization tandem mass spectrometry (HPLC/ESI-MS[2]).

HGs were measured and identified following previously established analytical protocols[16,67]. Separation of HGs was achieved using a Waters Alliance 2690 HPLC system equipped with a Phenomenex Luna $NH_2$ column ($150 \times 2\,mm$; $3\,\mu m$ particle size) and a guard column of the same material, which were both maintained at $30\,°C$. HGs were eluted using the following gradient profile: 95% A/5% B to 85% A/15% B in 10 min (held 7 min) at $0.5\,mL\,min^{-1}$, followed by backflushing with 30% A/70% B at $0.2\,mg\,mL^{-1}$ for 25 min and re-equilibrating the column with 95% A/5% B for 15 min. Solvent A was $n$-hexane:2-propanol:$HCO_2H$:14.8 M $NH_3$ aq. (79:20:0.12:0.04; v:v:v:v) and solvent B was 2-propanol:$H_2O$:$HCO_2H$:14.8 M $NH_3$ aq. (88:10:0.12:0.04; v:v:v:v). Detection of HGs was performed using a Micromass Quattro LC triple quadrupole MS equipped with an ESI interface and operated in positive ion mode. Source conditions were as follows: capillary 3.5 kV, cone 20 V, desolvation temperature $230\,°C$, source temperature $120\,°C$; cone gas $80\,L\,h^{-1}$ and desolvation gas $230\,L\,h^{-1}$. HGs were recorded in multiple reaction monitoring (MRM) mode using previously established mass spectral information[19,30,67] and monitoring the following transitions: $m/z\ 547 \rightarrow 415$ (pentose $HG_{26}$ diols), $m/z\ 561 \rightarrow 415$ (deoxyhexose $HG_{26}$ diols), $m/z\ 575 \rightarrow 413$ ($HG_{26}$ keto-ols), $m/z\ 577 \rightarrow 415$ ($HG_{26}$ diols), $m/z\ 603 \rightarrow 441$ ($HG_{28}$ keto-ols), $m/z\ 605 \rightarrow 443$ ($HG_{28}$ diols), $m/z\ 619 \rightarrow 457$ ($HG_{28}$ keto-diols), $m/z\ 621 \rightarrow 459$ ($HG_{28}$ triols), $m/z\ 631 \rightarrow 469$ ($HG_{30}$ keto-ols), $m/z\ 633 \rightarrow 471$ ($HG_{30}$ diols), $m/z\ 647 \rightarrow 485$ ($HG_{30}$ keto-diols), $m/z\ 649 \rightarrow 487$ ($HG_{30}$ triols), $m/z\ 675 \rightarrow 513$ ($HG_{32}$ keto-diols), and $m/z\ 677 \rightarrow 515$ ($HG_{32}$ triols). Quantification of HGs was achieved by integration of peak areas using the QuanLynx application software. To study changes in cyanobacterial community composition, fractional abundances (FA) of HGs were calculated. For this, the abundance of individual HGs was divided by the summed abundances of all HGs detected in a given sample.

In the East African and polar to tropical lakes, $HG_{26}$ diols and $HG_{26}$ keto-ols usually consisted of two structural isomers. The relative proportion of these components was calculated as the abundance of each isomer in relation to the abundance of all $HG_{26}$ diols and $HG_{26}$ keto-ols:

$$FA(HG_{26}x/y) = [HG_{26}keto-ol(I) + HG_{26}keto-ol(II) + HG_{26}diol(I) + HG_{26}diol(II)], \quad (1)$$

where $x$ denotes $HG_{26}$ keto-ol (I) or $HG_{26}$ keto-ol (II) and $y$ represents $HG_{26}$ diol (I) or $HG_{26}$ diol (II). Bivariate correlation analysis showed that SWT in these lakes was most significantly correlated with changes in the relative abundance of the most dominant of these isomers. Consequently, the most abundant structural isomer of the $HG_{26}$ diol and $HG_{26}$ keto-ol in each sample was used for the determination of the $HDI_{26}$:

$$HDI_{26} = HG_{26}diol/(HG_{26}keto-ol + HG_{26}diol) \quad (2)$$

**Statistical analyses**. In order to identify statistically significant correlations between HG distributions and the $HDI_{26}$ with environmental parameters, we performed two-tailed bivariate correlation analysis. The degree of association between environmental variables and fractional abundances of $HG_{26}$ keto-ols and $HG_{26}$ diols as well as the $HDI_{26}$ after removing the effect of SWT as controlling variable was tested using two-tailed partial correlation analysis. Comparison of slopes and intercepts between the regression of the East African and other calibration functions was achieved by analysis of covariance (ANCOVA). All of these analyses were performed using SPSS Statistics (Version 27.0). Hierarchical clustering of the East African lakes based on HG distribution patterns was achieved using the scientific data analysis software Past (Version 4.03). The chord distance, a modification of the Euclidean distance, was applied to the normalized data set as it provides robust results in ecological resemblance studies[68].

## Data availability

The authors declare that all data supporting the findings of this study are available within the paper and its Supplementary Information files. Source data are provided with this paper.

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

## Acknowledgements

Christoph Mayr provided surface sediments of Laguna Potrok Aike. Roger E. Summons and Ian Hawes are thanked for the provision of mat samples from Bratina Island, Antarctica. Martin Wessels and Sandra Böddeker are acknowledged for assistance during the Lake Constance sampling campaign. The environmental agency Baden-Württemberg is thanked for providing SWT data from Lake Constance for the time period from 2011 to 2015. We acknowledge assistance from Satria Bijaksana and the Institut Teknologi Bandung during field work in Indonesia. This research was funded by Christian-Albrechts-University through institutional grants awarded to L.S. Funding for T.W.E. was provided by the NASA Exobiology program (grant number: 80NSSC19K0465 awarded to Roger E. Summons) and also by the Alexander von Humboldt-Gesellschaft through the Feodor-Lynen-Fellowship.

## Author contributions

T.B. and L.S. designed the study. J.M.R., T.W.E. and A.S. provided sample material and environmental data. T.B. performed the analysis of HGs and interpretation of the results. L.S. provided personnel and laboratory facilities. The manuscript was written by T.B. All co-authors commented on the manuscript and provided input to its final version.

## Funding

## Competing interests

The authors declare no competing interests.
