## [Peer Review File · Nature Communications]

REVIEWER COMMENTS

Reviewer #1 (Remarks to the Author):

Bauersachs et al present a very compelling data set to calibrate the HDI26 paleotemperature proxy using surface sediments primarily from East Africa, and also from a few temperate and polar lakes. The data set shows a strong and significantly significant linear relationship between surface water temperature and the relative abundances of two glycolipids HG26 keto-ol and HG26 diol. For people interested in developing paleotemperature records from lacustrine sedimentary archives, this is a very exciting result and extends the existing toolkit of lipid-based paleothermometers. The study includes a full examination and clustering analysis of the distribution of heterocyte glycolipids detected in each lake in order to provide insight to the community structure of the cyanobacteria in each lake responsible for production. The authors also provide a downcore temperature reconstruction from Lake Tanganyika that can be compared with an existing GDGT-based paleotemperature record from the same lake (as well as to nearby Lake Malawi). The agreement in temperature histories from the two different approaches is very impressive, with the entirety of the ~40,000 yr temperature records agreeing within each of their confidence intervals. The paper is very well-written (with some grammatical errors throughout) and makes a compelling case for the utility of the HDI26 index for continental paleotemperature reconstruction.

That being said, I am not convinced that the paper belongs in Nature Communications. The authors attempt to justify the broad appeal of the study by beginning the paper with a long introduction concerning the importance of paleoclimate reconstructions for validating climate models, and therefore for improving future climate projections. This is all true, and I do not argue with the premise, but the results from this study are not actually used in that manner in this study. Because of this, the introduction sets up expectations for something "larger" than is delivered. The results and the beautiful data set presented do support the conclusions reached by this paper. However, those conclusions are that there is a strong relationship between production temperature and the HDI26 index, and that HG distributions apparently reflect variations in ecosystem types. I do not think that these conclusions are of broad enough interest for publication in Nature Communications. These exciting and compelling results would be more suitable for a more specific journal, such as GCA.

Comments:

What is the explanation for the different isomers? Is one a degradation product or are they interpreted as having been produced in the fractional abundances that they are observed in the sediments? If the former, then the inclusion of only the dominant isomer seems problematic. Has an attempt been made to construct the HDI26 index using a version of the index that sums the two isomers rather than discarding the smaller of the two for both the HG26 diol and HG26 keto-ol? It strikes me as odd that only the dominant isomer would be included, especially when it seems that the isomers are quite close in concentration in many (or at least in some) samples. Would the approach be improved, or more widely applicable if the isomers were summed?

The study concerns two aspects of HGs. One is the temperature sensitivity of the HG26 molecules. The other is the biozonation that is apparent in the full distribution patterns. However, there is very little discussion as to whether or not these two observations are connected. Is the HDI26 index picking up on speciation changes that are correlated with growth temperature? The fact that previous growth experiments with monocultures show a temperature dependence argues that there is a direct temperature effect on the index. Yet, the changing distributions that indicate changing community structures would be expected to complicate a simple HDI-temperature calibration. For instance, if the biozone 1 community composition is grown down to temperatures experienced by the higher altitude biozone 3, are the HDI values expected to be the same as observed in the biozone 3 lakes, or are those values dependent on the community structure shift?

Another factor that isn't discussed is the UV environment of the different sites. Certain microbes change their lipid production in response to light environment, and the authors allude to this in the discussion of "shading" by cyanobacterial mats - however, they discuss this more in terms of the thermal impact of shading. Is there any reason to expect that light intensity itself can impact the distribution patterns or keto-ol vs diol abundances?

I do not think the discussion about the "offset" between TEX86 and HDI26 in Tanganyika's Holocene sediments is really warranted. The 1.6 °C offset is easily within the calibration uncertainties of the proxies. Speculation about factors leading to the offset seems unnecessary and isn't very convincing.

The calibration applied to the Tanganyika record is one that spans the full modern distribution of community structures represented in the surface sediment transect - three different biozones and a temperature range from 5-27°C. Would it make sense to construct a calibration that only consists of the distributions that are observed in the Tanganyika record? The temperature variations experienced in Tang are only ~22-26 °C and therefore the full range of the modern calibration is not needed and might actually be skewing the inferred temperature relationship away from reality due to inclusion of a community of cyanobacteria that were never present in Tanganyika.

My overall thoughts are that this is an excellent study and will be of interest to organic geochemists and a subset of paleoclimatologists. Nature Communications does not seem to be the appropriate journal for this study. The above comments are meant to be constructive and to modestly improve what I consider to be a very well-written and compelling manuscript.

Reviewer #2 (Remarks to the Author):

Dear editor, dear authors,

With great interest I have read and reviewed the manuscript of Thorsten Bauersachs et al., entitled "A heterocyte glycolipid-based calibration to reconstruct past continental climate change". The manuscript is very well written and the science well described and concise. All the questions that should be answered, that is, the what, the why and the how, are addressed in a thorough and for the reader understandable manner.

This study describes a novel way to reconstruct the surface water temperatures of freshwater environments, and show that heterocyte glycolipids derived from cyanobacteria, can be applied to climate systems that date back to the last glaciation. That this organic proxy has potential for reconstructions up to the late Cretaceous is very promising.

The need for such proxies is very important for improving our knowledge of the climate systems of the past and with this information to improve climate models to predict future climate change more accurately. This makes this research valuable to various fields of climatology.

I therefore suggest publishing this manuscript after only minor changes.

My comments, suggestions and questions:

Line 16 in Abstract "... In tropical East African lakes and other polar to tropical freshwater environments"

I understand that you want to emphasize the ubiquitous distribution of heterocytous cyanobacteria. When first reading this sentence, I was however put off by the word placement of the word "other". I suggest to rephrasing this sentence to "... In tropical East African lakes and polar to other tropical freshwater environments"

Line 49

Change: "To this end, organic (LDI, UK'37, TEX86) and inorganic palaeothermometers ($\delta^{18}O$,

Mg/Ca)", Into: "To this end, organic (LDI, UK'37, TEX86) and inorganic ($\delta^{18}\text{O}$, Mg/Ca) palaeothermometers"

Line 226 – 234 and 277-294

I was wondering if there is a link between the optimal growth temperatures of the cyanobacteria and the N₂ fixation. As you discuss, optimum growth appears to be in Summer when temperatures are high. Furthermore, you say that the structural variation in HG's is controlled by the lake surface temperature mainly to make the N₂ fixation optimal. Does this mean that the cyanobacteria can keep this N₂ fixation to a constant optimum or is does this still occur at the preferred Summer temperatures?

Line 312-314

I'm very intrigued by this process and especially that there is evidence from remote sensing. However, you are citing an article from 1993 that in their method use AVHRR satellite images. Is this the only research on this subject? I know that Sentinel-3 is frequently used to study algal and cyanobacterial blooms. This satellite carries a SLSTR instrument, which monitors sea surface temperatures, and is more accurate and advanced then the AVHRR sensors used in the cited study.

Another question on this subject, and in line with my previous question on N₂ fixation. Does this process of increased surround temperature also increase their N₂-fixation rates? If so, then this seems quite important for future climate change and has consequences for the nitrogen cycle. Because of this, I believe this requires a bit more attention in your discussion.

Reviewer #3 (Remarks to the Author):

Summary

In this manuscript Bauersachs et al. expand on previous work on heterocyte glycolipids (HGs) by the lead author and develop a new lacustrine organic paleothermometer. For this purpose the authors analyse the African lake sediment dataset for its HG content, but also explore the HG distribution in a few lakes from other continents and climate zones. The authors develop a lake calibration based on the HDI26 index and then apply that to a sediment core from Lake Tanganyika that spans the last 37 kyr to test the application of their proxy and calibration. Although some inconsistencies exist, overall the HDI26 performs well and results in the expected climatic trends with a cold glacial and deglacial warming.

Main assessment

This is a clearly written manuscript that was easy to read and provides novel data. It is a classic calibration paper that I believe will be valuable for years to come to the organic geochemistry and paleoclimate communities. The data is technically sound and the data support their conclusions. Although I am sure many hurdles will come up, this preliminary work suggests that the HDI26 index has a promising future as paleoclimate proxy. I have a number of comments/suggestions that the authors need to address before publication, but overall I want to congratulate the authors with an excellent piece of work.

David Naafs

General comments

1. I would like to see a more thorough discussion of what is driving the temperature dependence of their proxy. Is it community change (as suggested by the hierarchical clustering?) or active membrane adaptation in response to temperature by cyanobacteria? Or a mixture of both? It does not mean that their proxy doesn't work, but it would be nice to see a more expanded discussion of the underlying mechanism(s).

2. In addition, the authors explicitly assume that all HGs are produced in the water column, but I didn't find clear evidence to support this claim. Could some (most for small lakes?) of the HGs in their lacustrine sediments be produced in the surrounding soils and then washed into the lake? These soil cyanobacteria likely experienced the same temperature as the lake, so it does not mean we can't use the HD26 index anymore, but the source of HGs and potential for a soil input needs to be discussed somewhere in the manuscript.

3. I don't really understand the potentially worrying statements in lines 257-261. The authors state that changes in cyanobacterial community might require regional calibrations. But why is that? The authors show that in the African dataset there are major changes in cyanobacterial community, but despite this (or maybe because of this, see comment 1 above) the HDI26 index displays a linear correlation with temperature and this correlation is pretty consistent with that found in other lakes (figure 5). The slope might be slightly different from that with lake Schreventeich, but given the scatter in the data set that is likely not significant.

4. Although the African lake dataset comprises the majority of the dataset, the authors do provide additional lacustrine samples, which is great. However, I wonder why these samples were not included in the hierarchical clustering (e.g. Figure 3). For example, the "temperate" lake samples should plot together with African lakes of similar temperature, right?

5. I appreciate that for figure 5 the authors plot all lake data together. There appears to be a high level of consistency between the different sample sets, which is great and re-assuring (no "local" lake effects it seems). But I then wonder why the authors do not generate a global calibration but opt for a local African lake calibration. Would a global calibration not be more valuable? Any reason why a local calibration is needed (besides that the R2 is likely slightly better for a local calibration)? For paleoclimate application I think global calibrations that sample all modern environments is much more robust.

6. The authors cite a number of previous papers that looked at the HG distribution in freshwater environments and cultures (lines 193-199). It would be great to add that data to figure 5 and determine whether all this published data is consistent with their findings and temperature dependence. If this published data differs, the authors need to explore why.

Minor comments

Maybe I missed something, but I could only find supplementary figure 1. Not sure what happened to the other supplementary figures.

I suggest the authors change the beginning of the introduction. This introduction is not terribly relevant for a calibration paper and in some instances appears to be incorrect. For example, the difference in temperature response between RCP 2.6 and 8.5 is proposed to reflect "inconsistent projections of the future global warming" (line 28-30), but of course the different RCP scenarios have very different forcings. Delete lines 24-46 and instead provide a more thorough introduction into the history of (organic) temperature proxies, especially those that are applied to lakes (e.g. TEX86 and MBT'5me).

Line 116: add "distinct HG distributions"

Lines 193-199

Line 368: with such a low resolution record I would not discuss millennial-scale variability such as the 8.2 kyr event.

REVIEWER COMMENTS

Reviewer #1 (Remarks to the Author):

Bauersachs et al present a very compelling data set to calibrate the HDI₂₆ paleotemperature proxy using surface sediments primarily from East Africa, and also from a few temperate and polar lakes. The data set shows a strong and significantly significant linear relationship between surface water temperature and the relative abundances of two glycolipids HG₂₆ keto-ol and HG₂₆ diol. For people interested in developing paleotemperature records from lacustrine sedimentary archives, this is a very exciting result and extends the existing toolkit of lipid-based paleothermometers. The study includes a full examination and clustering analysis of the distribution of heterocyte glycolipids detected in each lake in order to provide insight to the community structure of the cyanobacteria in each lake responsible for production. The authors also provide a downcore temperature reconstruction from Lake Tanganyika that can be compared with an existing GDGT-based paleotemperature record from the same lake (as well as to nearby Lake Malawi). The agreement in temperature histories from the two different approaches is very impressive, with the entirety of the ~40,000 yr temperature records agreeing within each of their confidence intervals. The paper is very well-written (with some grammatical errors throughout) and makes a compelling case for the utility of the HDI₂₆ index for continental paleotemperature reconstruction.

That being said, I am not convinced that the paper belongs in Nature Communications. The authors attempt to justify the broad appeal of the study by beginning the paper with a long introduction concerning the importance of paleoclimate reconstructions for validating climate models, and therefore for improving future climate projections. This is all true, and I do not argue with the premise, but the results from this study are not actually used in that manner in this study. Because of this, the introduction sets up expectations for something “larger” than is delivered. The results and the beautiful data set presented do support the conclusions reached by this paper. However, those conclusions are that there is a strong relationship between production temperature and the HDI₂₆ index, and that HG distributions apparently reflect variations in ecosystem types. I do not think that these conclusions are of broad enough interest for publication in Nature Communications. These exciting and compelling results would be more suitable for a more specific journal, such as GCA.

We are convinced that introducing a novel and independent organic temperature proxy that allows high-resolution continental climate studies is indeed of broad appeal to the scientific community as well as stakeholders and policymakers. The focus of the introduction, however, may have been too general. Therefore, we now significantly shortened the introduction and following Reviewer 3's suggestion placed a stronger emphasis on the history of organic temperature proxies in palaeoclimate research (ll. 24-41).

Comments:

What is the explanation for the different isomers? Is one a degradation product or are they interpreted as having been produced in the fractional abundances that they are observed in the sediments? If the former, then the inclusion of only the dominant isomer seems problematic.

Previous culture experiments (Bauersachs et al., 2019; Miller et al., 2020) and ongoing work conducted in our laboratory demonstrates that HGs may consist of up to four structural isomers in cyanobacterial isolates. These may derive from (1) variations in the type and stereochemistry of the sugar headgroup (Gambacorta et al., 1999) and (2) the position of the functional groups at the side

chain (Gambacorta et al., 1998; Bauersachs et al., 2009b). Retention times of HGs found in sediments and cultured cyanobacteria are indeed indistinguishable from each other and thereby compelling evidence for the biological origin of HG isomers in sediments is provided. Contributions of HG isomers derived from diagenetic reactions, which may confound the robust application of the HDI₂₆, are thus unlikely. The reviewer, however, addressed a very relevant point that is not sufficiently covered in the manuscript and we now include a brief discussion on the origin of heterocyte glycolipids in sediment sequences (ll. 219-224).

Has an attempt been made to construct the HDI₂₆ index using a version of the index that sums the two isomers rather than discarding the smaller of the two for both the HG₂₆ diol and HG₂₆ keto-ol? It strikes me as odd that only the dominant isomer would be included, especially when it seems that the isomers are quite close in concentration in many (or at least in some) samples. Would the approach be improved, or more widely applicable if the isomers were summed?

We initially devised several modifications of the HDI₂₆ using (1) only the most abundant HG isomers, (2) different mixtures of HG₂₆ diol and HG₂₆ keto-ol isomers and (3) the summed abundance of HG₂₆ diol and HG₂₆ keto-ol isomers. Linear regression analysis indicates that most iterations of the HDI₂₆ are strongly correlated with surface water temperatures. This is particularly the case for versions of the proxy that include the most abundant isomers as well as for the summed abundance of all isomers. In both variations, the HDI₂₆ is significantly correlated with surface water temperature in the set of East African lakes (most dominant isomers = $p < 0.001$ and $r^2 = 0.95$; all isomers = $p < 0.001$ and $r^2 = 0.95$) with the difference of the slope of both regression lines being statistically insignificant. Including all isomers in the calculation of the HDI₂₆ in the subpolar and polar lakes, however, results in a larger overall scatter and an increase in the residual error of the temperature calculation. In our opinion, this may result either from a different temperature adaption in the synthesis of HG isomers at lower temperatures that overproportionally affects the robust calculation of the HDI₂₆ in (sub)polar settings or the larger seasonality complicates the calibration of the HDI₂₆ with temperature. Therefore, we conclude that by only including the most abundant isomers in the calculation of the HDI₂₆, the most robust temperature reconstruction on a global scale can be achieved. We acknowledge that the difference between both iterations of the HDI₂₆ is only minor and now indicate that on a global scale the use of the most dominant HG₂₆ diol and HG₂₆ keto-ol isomers is considered to yield the most robust temperature reconstruction (ll. 258-260).

The study concerns two aspects of HGs. One is the temperature sensitivity of the HG₂₆ molecules. The other is the biozonation that is apparent in the full distribution patterns. However, there is very little discussion as to whether or not these two observations are connected. Is the HDI₂₆ index picking up on speciation changes that are correlated with growth temperature? The fact that previous growth experiments with monocultures show a temperature dependence argues that there is a direct temperature effect on the index. Yet, the changing distributions that indicate changing community structures would be expected to complicate a simple HDI-temperature calibration. For instance, if the biozone 1 community composition is grown down to temperatures experienced by the higher altitude biozone 3, are the HDI values expected to be the same as observed in the biozone 3 lakes, or are those values dependent on the community structure shift?

In a previous culture experiment, we were able to demonstrate that the HDI₂₆ may differ by 0.1 units in two cyanobacterial isolates grown at the same temperature (Bauersachs et al., 2014). Therefore, we fully agree with the reviewer that changes in the cyanobacterial community composition may potentially contribute to the HDI₂₆ and therefore complicate the application of the HDI₂₆ in palaeotemperature reconstructions. In lake environments, however, heterocyte glycolipids do not

only derive from one single species and the sedimentary HG signal represents a mean of the individual contributions from multiple cyanobacterial groups integrated over several years to decades. Species effects are thus - to a large extent - averaged out and as such they are not expected to overly impact the HDI₂₆. Moreover, although our data suggests that there is a shift in the cyanobacterial community along the altitudinal transect (through changes in the relative proportion of HG species, which are representative for individual families and genera), this does not imply that the biological sources of HG₂₆ diols and HG₂₆ keto-ols are necessarily different in lakes of Biozones 1-3. The strong linear correlation observed in our data set indeed argues against a major species-dependent effect on the calculation of the HDI₂₆ as this is assumed to result in a significantly larger scatter and residual error in the data set. However, some of the minor offsets observed in the HDI₂₆ calibration may indeed result from changes in the cyanobacterial community and this issue is now more thoroughly discussed in the manuscript (ll. 245-254).

Another factor that isn't discussed is the UV environment of the different sites. Certain microbes change their lipid production in response to light environment, and the authors allude to this in the discussion of "shading" by cyanobacterial mats - however, they discuss this more in terms of the thermal impact of shading. Is there any reason to expect that light intensity itself can impact the distribution patterns or keto-ol vs diol abundances?

We do not want to exclude that light intensity can impact HG distribution patterns or the ratio of HG₂₆ diols vs HG₂₆ keto-ols but at present there is no evidence for this. To date, temperature has been shown to strongly affect the synthesis of heterocyte glycolipids in cultures of cyanobacteria (Bauersachs et al., 2009a; 2014) and lake surface waters (Wörmer et al., 2012). Bauersachs et al. (2015) demonstrated that lake surface water temperature strongly correlates with the relative abundance of HG₂₆ diols and HG₂₆ keto-ols, while neither oxygen concentration nor pH did show a significant correlation with HG distribution patterns or the relative abundance of individual HGs. The effect of environmental parameters other than those mentioned above on HG distribution patterns and the relative abundances of HGs, however, has not been studied yet. Given that the chemical structure of heterocyte glycolipids (i.e. lack of unsaturation and aromatic ring structures) precludes their effective use as antioxidants, it is unlikely though that light intensity or changes in the intensity of UV radiation significantly impact the synthesis of HGs. We agree that future studies will contribute to determine whether environmental factors other than temperature (also including light intensity) may affect the distribution and abundance of HGs in environmental samples and state this in the discussion (ll. 322-325).

I do not think the discussion about the "offset" between TEX₈₆ and HDI₂₆ in Tanganyika's Holocene sediments is really warranted. The 1.6 °C offset is easily within the calibration uncertainties of the proxies. Speculation about factors leading to the offset seems unnecessary and isn't very convincing.

We certainly agree with the reviewer that the temperature offset between the HDI₂₆ and the TEX₈₆ in the Holocene sediments is within the uncertainty range of both calibrations. However, we are also convinced that this temperature offset is not a simple artefact of the transfer functions as (1) the offset does not occur in Pleistocene sediments and (2) is by far too consistent in the late Holocene sediment sequence to be a calibration uncertainty. The latter would be expected to show deviations into both directions, which is not the case. Therefore, we feel that a brief discussion of possible factors explaining the offset between both proxies is warranted and of benefit for the manuscript.

The calibration applied to the Tanganyika record is one that spans the full modern distribution of community structures represented in the surface sediment transect - three different biozones and a

temperature range from 5-27°C. Would it make sense to construct a calibration that only consists of the distributions that are observed in the Tanganyika record? The temperature variations experienced in Tang are only ~22-26 °C and therefore the full range of the modern calibration is not needed and might actually be skewing the inferred temperature relationship away from reality due to inclusion of a community of cyanobacteria that were never present in Tanganyika.

We appreciate the issue raised by the reviewer. A calibration (hereafter referred to as “Lake Tanganyika calibration”) that only includes lake surface sediments falling within the temperature range of modern Lake Tanganyika, however, is considerably weaker ($r^2 = 0.34$; $n = 23$) compared to the originally proposed East African lake calibration ($r^2 = 0.95$, $n = 42$). Surface water temperatures reconstructed using the “Lake Tanganyika calibration” range from 18.3 °C during the last glacial maximum to 29.6 °C in the modern lake and evidence deglacial warming of >11 °C. Such a value is beyond any reconstructed low-altitude deglacial warming in tropical East Africa, which based on lipid palaeothermometers (Powers, 2005; Tierney et al., 2008) and vegetation changes (Bonnefille et al., 1990; Farrera et al., 1999) is estimated to vary between 2 to 4 °C. Such a temperature range agrees with the 4.1 °C warming inferred from the HDI₂₆ and using our “East African lake calibration”. The “Lake Tanganyika calibration” thus does not seem to yield reasonable temperature estimates. Moreover, we do not feel that establishing a temperature calibration that only covers a very narrow interval at the upper end of the temperature spectrum, is an appropriate approach as the magnitude of climate variations in a palaeoenvironmental context cannot be foreseen and may be much larger than the 4-5 °C temperature difference covered by the “Lake Tanganyika calibration”. We are thus grateful for the reviewer’s suggestion but as such a calibration does not result in improved temperature reconstructions, it is not further discussed in the manuscript.

My overall thoughts are that this is an excellent study and will be of interest to organic geochemists and a subset of paleoclimatologists. Nature Communications does not seem to be the appropriate journal for this study. The above comments are meant to be constructive and to modestly improve what I consider to be a very well-written and compelling manuscript.

Reviewer #2 (Remarks to the Author):

Dear editor, dear authors,

With great interest I have read and reviewed the manuscript of Thorsten Bauersachs et al., entitled "A heterocyte glycolipid-based calibration to reconstruct past continental climate change". The manuscript is very well written and the science well described and concise. All the questions that should be answered, that is, the what, the why and the how, are addressed in a thorough and for the reader understandable manner.

This study describes a novel way to reconstruct the surface water temperatures of freshwater environments, and show that heterocyte glycolipids derived from cyanobacteria, can be applied to climate systems that date back to the last glaciation. That this organic proxy has potential for reconstructions up to the late Cretaceous is very promising.

The need for such proxies is very important for improving our knowledge of the climate systems of the past and with this information to improve climate models to predict future climate change more accurately. This makes this research valuable to various fields of climatology. I therefore suggest publishing this manuscript after only minor changes.

My comments, suggestions and questions:

Line 16 in Abstract "... In tropical East African lakes and other polar to tropical freshwater environments". I understand that you want to emphasize the ubiquitous distribution of heterocytous cyanobacteria. When first reading this sentence, I was however put off by the word placement of the word "other". I suggest to rephrasing this sentence to "... In tropical East African lakes and polar to other tropical freshwater environments"

We acknowledge that the wording of the sentence is somewhat ambiguous and very much thank the reviewer for her/his suggestion for improving the sentence. We think, however, that the following modification "...in equatorial East African lakes as well as polar to tropical freshwater environments" is more elegant and we changed the sentence accordingly.

Line 49: Change: "To this end, organic (LDI, U^{K}_{37} , TEX_{86}) and inorganic palaeothermometers ($\delta^{18}O$, Mg/Ca)", Into: "To this end, organic (LDI, $U^{K}37$, TEX_{86}) and inorganic ($\delta^{18}O$, Mg/Ca) palaeothermometers"

Due to the major changes made in the introduction, this modification is not applicable anymore.

Line 226 – 234 and 277-294: I was wondering if there is a link between the optimal growth temperatures of the cyanobacteria and the N_2 fixation. As you discuss, optimum growth appears to be in Summer when temperatures are high. Furthermore, you say that the structural variation in HGs is controlled by the lake surface temperature mainly to make the N_2 fixation optimal. Does this mean that the cyanobacteria can keep this N_2 fixation to a constant optimum or is does this still occur at the preferred Summer temperatures?

Nitrogenase, the enzyme needed for biological N_2 fixation, is oxygen sensitive. Therefore, it has to operate in an anaerobic environment for which the presence of a gas diffusion barrier, consisting of heterocyte glycolipids, is essential. We would like to stress, however, that achieving a low

permeability of the gas diffusion barrier does not automatically result in highest N₂ fixation rates because N₂ fixation requires energy and as such is highest during summer when a maximum in light intensity is observed. This essentially implies that cyanobacteria have the ability to ensure optimal N₂ fixation rates at any given time of the year by adjusting the composition of the gas diffusion barrier but that external factors (such as e.g. light intensity and the availability of trace metals required for the synthesis of nitrogenase) control the time period of highest N₂ fixation. Although an interesting subject by itself, a more detailed discussion on N₂ fixation by cyanobacteria seems to be beyond the scope of the manuscript and does not contribute to the development of the HDI₂₆ as proxy in palaeoclimate research.

Line 312-314: I'm very intrigued by this process and especially that there is evidence from remote sensing. However, you are citing an article from 1993 that in their method use AVHRR satellite images. Is this the only research on this subject? I know that Sentinel-3 is frequently used to study algal and cyanobacterial blooms. This satellite carries a SLSTR instrument, which monitors sea surface temperatures, and is more accurate and advanced than the AVHRR sensors used in the cited study.

As pointed out by the reviewer, remote sensing is a valuable and well-established tool for detecting and mapping the spatiotemporal extent of algal as well as cyanobacterial blooms. Studies investigating local heating effects of cyanobacteria on surface waters, however, are sparse and data provided by the "Sea and Land Surface Temperature Radiometer (SLSTR)" installed on Sentinel 3 has, to the best of our knowledge and after a careful literature research, not been used yet to evaluate the impact of cyanobacteria on local to regional water temperatures. This may relate to the fact that the two Sentinel-3 systems have only been launched in 2016 and 2018 and as such have rather recently commenced operations. Accordingly, studies tackling the effect of biologically-mediated heating of surface waters are expected to be available in the future but have not been published yet. After additional literature research, however, we note that the extent of local surface water heating by cyanobacteria may be more variable and range from ~1 °C to ~5 °C based on in-situ measurements (Wurl et al., 2018) and remote sensing with the "Advanced Very High-Resolution Radiometer (AVHRR)" sensor on board the NOAA-12 and NOAA-14 satellites (Capone et al. 1998). This observed temperature range, in particular towards the high end, agrees well with the too high temperatures predicted by the HDI₂₆ in some of the East African high elevation tarns. This supports our argumentation of a local heating effect due to the absorption of light energy by cyanobacterial pigments. We now included both references above in our study and point to the need of further studies to elucidate the effect of energy absorption and water column heating on the application of the HDI₂₆ (ll. 322-325).

Another question on this subject, and in line with my previous question on N₂ fixation. Does this process of increased surround temperature also increase their N₂-fixation rates? If so, then this seems quite important for future climate change and has consequences for the nitrogen cycle. Because of this, I believe this requires a bit more attention in your discussion.

We agree that the possible feedback of global warming on cyanobacterial N₂ fixation is a fascinating and very relevant topic. Previous cultures studies have demonstrated that cyanobacteria express optimum N₂ fixation rates at growth temperatures between 24 to 30 °C (see Paerl and Otten (2013) and reference therein). This implies that global warming will lead to increased cyanobacterial N₂ fixation in the future, which in turn will affect primary productivity as well as biogeochemical cycles and hence is of relevance for the global carbon and nitrogen budget. This is - without doubt - an exciting topic. At the same time, however, we feel that this topic is far beyond the scope of the manuscript, which is the development and application of a novel lipid palaeothermometer to

reconstruct past continental climate change. In order to keep the focus on the temperature proxy, we hence did not add a discussion on the effect of global warming on cyanobacterial N_2 -fixation to the manuscript.

Reviewer #3 (Remarks to the Author):

Summary

In this manuscript Bauersachs et al. expand on previous work on heterocycle glycolipids (HGs) by the lead author and develop a new lacustrine organic paleothermometer. For this purpose, the authors analyse the African lake sediment dataset for its HG content, but also explore the HG distribution in a few lakes from other continents and climate zones. The authors develop a lake calibration based on the HDI₂₆ index and then apply that to a sediment core from Lake Tanganyika that spans the last 37 kyr to test the application of their proxy and calibration. Although some inconsistencies exist, overall the HDI₂₆ performs well and results in the expected climatic trends with a cold glacial and deglacial warming.

Main assessment

This is a clearly written manuscript that was easy to read and provides novel data. It is a classic calibration paper that I believe will be valuable for years to come to the organic geochemistry and paleoclimate communities. The data is technically sound and the data supports their conclusions. Although I am sure many hurdles will come up, this preliminary work suggests that the HDI₂₆ index has a promising future as paleoclimate proxy. I have a number of comments/suggestions that the authors need to address before publication, but overall I want to congratulate the authors with an excellent piece of work.

David Naafs

General comments

1. I would like to see a more thorough discussion of what is driving the temperature dependence of their proxy. Is it community change (as suggested by the hierarchical clustering?) or active membrane adaptation in response to temperature by cyanobacteria? Or a mixture of both? It does not mean that their proxy doesn't work, but it would be nice to see a more expanded discussion of the underlying mechanism(s).

Previous culture experiments have demonstrated that growth temperature is the main factor controlling the relative abundance of HG₂₆ diols and HG₂₆ keto-ols and thereby HDI₂₆ values in a given cyanobacterium (Bauersachs et al., 2009a; Bauersachs et al., 2014). In contrast, the community composition primarily determines the overall HG distribution pattern in a given lake system and whether HG₂₆ diols and HG₂₆ keto-ols will be present or not. To clarify this aspect, we now added a more detailed discussion on the mechanisms that underlie the HDI₂₆ (ll. 238-242).

2. In addition, the authors explicitly assume that all HGs are produced in the water column, but I didn't find clear evidence to support this claim. Could some (most for small lakes?) of the HGs in their lacustrine sediments be produced in the surrounding soils and then washed into the lake? These soil cyanobacteria likely experienced the same temperature as the lake, so it does not mean we can't use the HD₂₆ index anymore, but the source of HGs and potential for a soil input needs to be discussed somewhere in the manuscript.

A contribution of HGs from sources surrounding the lakes was indeed not extensively discussed in the manuscript. One of the reasons for this is that HG distributions in terrestrial cyanobacteria and subsequent transport mechanisms have not been studied yet at all. However, our data and that of a previous study (Bauersachs et al., 2015) indicate that the sedimentary HG signal captures the signal of highest cyanobacterial in-lake productivity. Given that such productivity, in particular during bloom events, likely exceeds cyanobacterial productivity in soils by several orders of magnitude, the contribution of HGs from allochthonous sources is likely to be minor. The strong correlation of HDI₂₆ values and surface water temperature also argues against a significant contribution of HGs from allochthonous sources as variable runoff between lakes should lead to a larger scatter in our dataset, which is not the case. However, we acknowledge that minute amounts of HGs may derive from terrestrial sources and now list a loading of allochthonous HGs as a factor that may potentially impact HG distribution patterns in lakes (ll. 322-325).

3. I don't really understand the potentially worrying statements in lines 257-261. The authors state that changes in cyanobacterial community might require regional calibrations. But why is that? The authors show that in the African dataset there are major changes in cyanobacterial community, but despite this (or maybe because of this, see comment 1 above) the HDI₂₆ index displays a linear correlation with temperature and this correlation is pretty consistent with that found in other lakes (figure 5). The slope might be slightly different from that with lake Schreventeich, but given the scatter in the data set that is likely not significant.

Comparison of the East African lake calibration with the global lake calibration indicates that although a similar slope of the regression is observed, absolute values based on both calibrations are different with the former yielding generally higher HDI₂₆ values. This finding suggests that although the HDI₂₆ is applicable on a global scale, there may be regional differences in the distributions of HGs in lake environments, in particular towards the lower end of the temperature spectrum. Therefore, we feel the necessity to point out that regional calibrations may indeed be needed for the robust reconstruction of past continental climates as stated in the manuscript (ll. 262-268).

4. Although the African lake dataset comprises the majority of the dataset, the authors do provide additional lacustrine samples, which is great. However, I wonder why these samples were not included in the hierarchical clustering (e.g. Figure 3). For example, the "temperate" lake samples should plot together with African lakes of similar temperature, right?

This does not necessarily have to be the case because the clustering is primarily controlled by the relative abundance of HGs with different chain lengths, which largely depends on the composition of the cyanobacterial community. As this community may differ on global scales and different cyanobacterial families may occur in lakes of tropical to polar climates, we choose to only include HG distribution patterns of the East African lakes in the cluster analysis because this allows an unbiased discussion of altitude-driven community shifts in a regionally confined dataset. Including the globally distributed lakes in the hierarchical clustering indeed results in tropical and temperate lakes to plot in Biozones 1 and 2, respectively. However, samples of Antarctic freshwater ponds also plot together with lakes of Biozone 2 due to the overall stronger influence of the community composition than altitude. Therefore, we conclude that for the purpose of discussing regional differences in the cyanobacterial community it is not beneficial to include lakes from other climate zones in the clustering.

5. I appreciate that for figure 5 the authors plot all lake data together. There appears to be a high level of consistency between the different sample sets, which is great and re-assuring (no "local" lake

effects it seems). But I then wonder why the authors do not generate a global calibration but opt for a local African lake calibration. Would a global calibration not be more valuable? Any reason why a local calibration is needed (besides that the R2 is likely slightly better for a local calibration)? For paleoclimate application I think global calibrations that sample all modern environments is much more robust.

We agree that a global calibration is ultimately desired. At present, however, we think that the number of lakes studied on a global scale is not extensive enough and that additional studies investigating the distribution of HG₂₆ diols and HG₂₆ keto-ols in a large number of lake surface sediments worldwide are needed to allow for the development of a statistically robust global calibration. Comparison of the East African lake calibration with the global lake calibration also indicates that the former generally yields higher HDI₂₆ values for a given temperature and that there is a consistent temperature offset between both calibration lines (Figure 5). This observation argues against merging both datasets and suggests that there is indeed a need for regional calibrations as well as for more studies that will constrain the response of HGs to different climatic conditions (ll. 259-267). We thus conclude that at this point establishing a global calibration is premature but will be subject of future investigations, for which this study will lie the foundation.

6. The authors cite a number of previous papers that looked at the HG distribution in freshwater environments and cultures (lines 193-199). It would be great to add that data to figure 5 and determine whether all this published data is consistent with their findings and temperature dependence. If this published data differs, the authors need to explore why.

We agree with the reviewer that it would be tempting to add data on HG distributions in freshwater environments and cultures to Figure 5. There are, however, multiple reasons why this is either not feasible and/or we consciously chose not to plot the data while drafting the manuscript. First, the lack of quantitative data regarding HG distributions and temperature data in previous publications. The majority of earlier literature on HGs reports the presence of these components in relative terms [(+) abundant; (o) average abundance; (-) minor abundance or not present] but no quantitative data in form of area counts, fractional abundances or percentages are provided. Hence, no HDI₂₆ value can be calculated. This includes the reference to Spanish freshwater lakes and Antarctic biofilms (Wörmer et al., 2012) as well as HG distributions reported from microbial biofilms of Svalbard (Rethemeyer et al., 2010) and HG distribution in cultured cyanobacteria (Bauersachs et al., 2009a; Wörmer et al., 2012). Second, outdated analytical techniques used for the detection of HGs prevents a direct comparison of datasets. HGs were initially eluted on DIOL columns, which do not allow separating the structural isomers that we describe here. In most previous publications, such as the one on HG distributions in North American lakes (Bale et al., 2016), only one structural isomer for each HG is reported. Given the importance of isomer separation for the correct calculating of the HDI₂₆, however, this introduces a none quantifiable error and HDI₂₆ values might be flawed under these circumstances. Third, if a microbial community exclusively consists of cyanobacteria other than Nostocaceae, as is the case in the Icelandic hot spring (Bauersachs et al., 2013), HG₂₆ diols and HG₂₆ keto-ols will not be present and HDI₂₆ values cannot be calculated. Although we are highly supportive of the reviewer's proposition, the simple lack of quantitative data and of data of sufficient analytical quality prevents us from plotting previously published HG distribution patterns in Figure 5 and also including them in a more elaborated discussion.

Minor comments

Maybe I missed something, but I could only find supplementary figure 1. Not sure what happened to the other supplementary figures.

We submitted and referred to only one supplementary figure in the manuscript (l. 55).

I suggest the authors change the beginning of the introduction. This introduction is not terribly relevant for a calibration paper and in some instances appears to be incorrect. For example, the difference in temperature response between RCP 2.6 and 8.5 is proposed to reflect “inconsistent projections of the future global warming” (line 28-30), but of course the different RCP scenarios have very different forcings. Delete lines 24-46 and instead provide a more thorough introduction into the history of (organic) temperature proxies, especially those that are applied to lakes (e.g. TEX₈₆ and MBT_{5Me}).

As a similar issue was also raised by Reviewer 1, we thoroughly revised the beginning of the introduction. Instead of focusing on various climate warming scenarios and uncertainties in future climate projections, we now provide more detailed information on the history of organic temperature proxies together with their value but also limitations in reconstructing past continental climate change (ll. 24-42).

Line 116: add “distinct HG distributions”

We have followed the reviewer’s suggestion and modified the sentence as follows: “...each characterized by distinct HG distributions and cyanobacterial communities as well as altitudinal ranges” (ll. 98-101).

Lines 193-199

The reviewer’s comment is apparently missing.

Line 368: with such a low-resolution record I would not discuss millennial-scale variability such as the 8.2 kyr event.

We agree that the record is not of highest resolution and given the uncertainty of the age model removed the discussion of the 8.2 kyr event as suggest by the reviewer.

References used in the response letter

- Bale N. J., Hopmans E. C., Schoon P. L., de Kluijver A., Downing J. A., Middelburg J. J., Sinninghe Damsté J. S. and Schouten S. (2016) Impact of trophic state on the distribution of intact polar lipids in surface waters of lakes. *Limnol. Oceanogr.* **61**, 1065–1077.
- Bauersachs T., Compaoré J., Hopmans E. C., Stal L. J., Schouten S. and Sinninghe Damsté J. S. (2009a) Distribution of heterocyst glycolipids in cyanobacteria. *Phytochemistry* **70**, 2034–2039.
- Bauersachs T., Hopmans E. C., Compaoré J., Stal L. J., Schouten S. and Sinninghe Damsté J. S. (2009b) Rapid analysis of long-chain glycolipids in heterocystous cyanobacteria using high-performance liquid chromatography coupled to electrospray ionization tandem mass spectrometry. *Rapid Commun. Mass Spectrom.* **23**, 1387–1394.
- Bauersachs T., Miller S. R., Gugger M., Mudimu O., Friedl T. and Schwark L. (2019) Heterocyte glycolipids indicate polyphyly of stigonematalean cyanobacteria. *Phytochemistry* **166**, 112059.
- Bauersachs T., Miller S. R., van der Meer M. T. J., Hopmans E. C., Schouten S. and Sinninghe Damsté J. S. (2013) Distribution of long chain heterocyst glycolipids in cultures of the thermophilic cyanobacterium *Mastigocladus laminosus* and a hot spring microbial mat. *Org. Geochem.* **56**, 19–24.
- Bauersachs T., Rochelmeier J. and Schwark L. (2015) Seasonal lake surface water temperature trends reflected by heterocyst glycolipid-based molecular thermometers. *Biogeosciences* **12**, 3741–3751.
- Bauersachs T., Stal L. J., Grego M. and Schwark L. (2014) Temperature induced changes in the heterocyst glycolipid composition of N₂ fixing heterocystous cyanobacteria. *Org. Geochem.* **69**, 98–105.
- Bonnefille R., Roeland J. C. and Guiot J. (1990) Temperature and rainfall estimates for the past 40,000 years in equatorial Africa. *Nature* **346**, 347–349.
- Farrera I., Harrison S. P., Prentice I. C., Ramstein G., Guiot J., Bartlein P. J., Bonnefille R., Bush M., Cramer W., Von Grafenstein U., Holmgren K., Hooghiemstra H., Hope G., Jolly D., Lauritzen S. E., Ono Y., Pinot S., Stute M. and Yu G. (1999) Tropical climates at the Last Glacial Maximum: A new synthesis of terrestrial palaeoclimate data. I. Vegetation, lake-levels and geochemistry. *Clim. Dyn.* **15**, 823–856.
- Gambacorta A., Pagnotta E., Romano I., Sodano G. and Trincone A. (1998) Heterocyst glycolipids from nitrogen-fixing cyanobacteria other than Nostocaceae. *Phytochemistry* **48**, 801–805.
- Gambacorta A., Trincone A., Soriente A. and Sodano G. (1999) Chemistry of glycolipids from the heterocysts of nitrogen-fixing cyanobacteria. *Curr. Top. Phytochem.* **2**, 145–150.
- Miller S. R., Longley R., Hutchins P. R. and Bauersachs T. (2020) Cellular Innovation of the Cyanobacterial Heterocyst by the Adaptive Loss of Plasticity. *Curr. Biol.* **30**, 344-350.e4.
- Paerl H. W. and Otten T. G. (2013) Harmful cyanobacterial blooms: Causes, consequences, and controls. *Microb. Ecol.* **65**, 995–1010.
- Powers L. A. (2005) Large temperature variability in the southern African tropics since the last glacial maximum. *Geophys. Res. Lett.* **32**, L08706.
- Rethemeyer J., Schubotz F., Talbot H. M., Cooke M. P., Hinrichs K. U. and Mollenhauer G. (2010) Distribution of polar membrane lipids in permafrost soils and sediments of a small high Arctic catchment. *Org. Geochem.* **41**, 1130–1145.
- Tierney J. E., Russell J. M., Huang Y., Sinninghe Damsté J. S., Hopmans E. C. and Cohen A. S. (2008) Northern hemisphere controls on tropical southeast African climate during the past 60,000 years. *Science* **322**, 252–255.
- Wörmer L., Cirés S., Velázquez D., Quesada A. and Hinrichs K.-U. (2012) Cyanobacterial heterocyst glycolipids in cultures and environmental samples: Diversity and biomarker potential. *Limnol. Oceanogr.* **57**, 1775–1788.

REVIEWERS' COMMENTS

Reviewer #1 (Remarks to the Author):

I have finished reviewing the revisions to Bauersachs et al. "A heterocyte glycolipid-based calibration to reconstruct past continental climate change" and I am satisfied with the revisions. Thanks to the authors for their careful considerations of not only my comments, but also the comments of the other two reviewers. I believe they have satisfactorily revised this manuscript and I have no further major comments.

Minor comment:

I am glad that the authors revised the introduction to make it more relevant to the study at hand. I will note however, that they did not do a very thorough job referencing the existing research on terrestrial biomarker-based proxies. There are a number of brGDGT studies that should really be cited, as well as a number of lacustrine alkenone studies. It would be appropriate to cite at least those that contain calibrations (global or local), in order to fully capture the large amount of effort that has gone into developing these terrestrial paleoclimate proxies. As it stands, the introduction under-represents these efforts.

Aside from that, I just want to say that this is an excellent study, and it is very well written. Congratulations to the authors on their fine work.

Reviewer #2 (Remarks to the Author):

I have read the revised submission by Bauersachs and colleagues. The authors have put a lot of effort into rebutting their manuscript. The manuscript has greatly improved and my scientific comments (as well as those by the other referees) have been addressed thoroughly and adequately.

I do not share the opinion with reviewer 1. I do believe this research is very valuable for climatology research and therefore has appeal to the wide audience of Nature Communications. I have no further comments; in my point of view the manuscript is ready for publication.

Reviewer #3 (Remarks to the Author):

I would like to thank the authors for providing a well-structured revised manuscript that addresses the majority of my original comments. However, I have a number of additional comments, listed below, that need to be addressed prior to publication.

David Naafs

Reply letter reply to point 2: although I follow the comment by the authors that soil input may not be a major source to most lakes, this caveat needs a bit more discussion in the main manuscript so that future readers are not left wondering whether soil input plays a major role. Can some of the reply be added to the main manuscript?

Reply letter reply to point 5: so here the authors argue that they only provide a local calibration, but this is nowhere mentioned in the abstract of the paper. In fact, the beginning of the current version of

the abstract argues that our understanding of continental climate is hampered by the lack of global applicable proxies and therefore hints at this paper providing an universal proxy. But this paper does not provide a global proxy.

To solve this, restructure your abstract (I suggest to delete sentence in line 13-14 (see also below)) and clearly state in the abstract that this is a local calibration for east African lakes only. Maybe even add this local calibration aspect to the title of the manuscript to avoid confusion and blind application of this calibration to other lakes by future studies.

Reply letter reply to point 6: I think this is a really good reply and this explanation for a lack of culture data to compare the surface sediment data with, in a condensed form, should be added to the main manuscript.

Line 13-14: delete this sentence as some universally applicable continental climate proxies exist (MBT⁵me to name one, but there are others). See also comment above.

Line 23: change to: "thereby quantifying past continental climate change."

Line 24: This is not correct. The temperature change (warming) we have so far experienced is at best 1.5 °C since the pre-industrial, even your deglacial temperature record shows more temperature change. I think you might confuse the extent of warming with the rate of warming, which MIGHT be unique?

Line 23-29: this beginning of the abstract still feels forced and not terribly relevant for a proxy calibration paper. I strongly suggest you delete this section and start with explaining the history or (organic) paleoclimate proxies.

Line 35 (and others): not sure what "punctuated" climate events are. Do you mean abrupt climate change events such as the Younger Dryas?

Line 100: can the authors add the altitude of each lake to figure 3? That would help to show the reader that the clustering fits with altitude.

Line 115-116: delete "widespread and"

Line 116: When I look at figure 3 it seems that HG26 diols and HG26 keto-ols have extremely low abundances in sample Kn?

Line 120+122: it is not clear to me how these "fractional abundances" were calculated. Can you give the equation for these two fractional abundances here so the reader understands what is discussed and shown in figure 4?

Line 141: delete "only"

Line 145: This lake has a latitude north of 60 °S, so technically not subpolar I guess but mid-latitudes?

Line 190-200: somewhere here the correlation with summer temperature and mean above freezing temperatures needs to be discussed. Why was SWT selected for the proxy? Although the seasonal cycle is small in these African lakes, this should become significant for the high-latitude lakes, as hinted at later in the discussion and as stated in the reply letter ("HG signal captures the signal of highest cyanobacterial in-lake productivity").

Line 243: also give RMSE here

Line 263-265: how can such community changes that impact the calibration be constrained in the past? How can we assess whether the temperature dependence has changed in down core records due to changes in the cyanobacterial community? This is a major issue that needs to be discussed.

Figure 5: Can you explain the meaning of the different coloured lines in figure 5? This is currently not clearly stated in the figure legend (e.g. what does the blue line mean, etc).

Line 305: although maybe not for TEX86, it is (maybe better) constrained for other organic proxies such as UK'37 and the LDI. Change sentence accordingly.

Line 355-356: something is wrong with this sentence. What does "keeping in with" mean?

Figure 6: This figure needs a bit more work. First, can you add the calibration RMSE for both TEX86 and the HDI25 calibrations to all the data points to better show the uncertainty associated with each record? Second, what does the error currently shown for HDI26 mean, can you state this in the caption? Third: delete the 8.2 kyr event

Line 364-375: delete this part. The temperature change during the YD is well within the RMSE of the calibration and based on 1 data point. Just focus on the long-term trends.

Line 382: This offset of 1.6 oC is well within the RMSE of both the TEX86 and HDI26 proxy, so are these two records really statistically different? That is why it would be good to add the RMSE for both these proxies to figure 6.

Data availability: prior to publication the data should also be uploaded to a public database such as Pangaea to secure future access.

REVIEWERS' COMMENTS

Reviewer #1 (Remarks to the Author):

I have finished reviewing the revisions to Bauersachs et al. "A heterocyte glycolipid-based calibration to reconstruct past continental climate change" and I am satisfied with the revisions. Thanks to the authors for their careful considerations of not only my comments, but also the comments of the other two reviewers. I believe they have satisfactorily revised this manuscript and I have no further major comments.

We are grateful for the reviewer's assessment and glad to hear that we satisfactorily addressed the previously expressed concerns.

Minor comment: I am glad that the authors revised the introduction to make it more relevant to the study at hand. I will note however, that they did not do a very thorough job referencing the existing research on terrestrial biomarker-based proxies. There are a number of brGDGT studies that should really be cited, as well as a number of lacustrine alkenone studies. It would be appropriate to cite at least those that contain calibrations (global or local), in order to fully capture the large amount of effort that has gone into developing these terrestrial paleoclimate proxies. As it stands, the introduction under-represents these efforts.

The focus of our introduction is placed on the lack of lipid palaeothermometers that are universally applicable in lacustrine environments. We did not include brGDGT and alkenone studies because the MBT'_{5Me} (based on brGDGTs) is a 'terrestrial' proxy considered to reflect mean annual air temperatures and the U'_{37} (based on alkenones) appears to be applicable in only some geographically constrained regions of our planet. In order to cover the full range of lipid palaeothermometers, we now also briefly refer to the aforementioned organic temperature proxies in the introduction.

Aside from that, I just want to say that this is an excellent study, and it is very well written. Congratulations to the authors on their fine work.

Reviewer #2 (Remarks to the Author):

I have read the revised submission by Bauersachs and colleagues. The authors have put a lot of effort into rebutting their manuscript. The manuscript has greatly improved and my scientific comments (as well as those by the other referees) have been addressed thoroughly and adequately.

I do not share the opinion with reviewer 1. I do believe this research is very valuable for climatology research and therefore has appeal to the wide audience of Nature Communications. I have no further comments; in my point of view the manuscript is ready for publication.

We appreciate the kind words of the reviewer.

Reviewer #3 (Remarks to the Author):

I would like to thank the authors for providing a well-structured revised manuscript that addresses the majority of my original comments. However, I have a number of additional comments, listed below, that need to be addressed prior to publication.

David Naafs

Reply letter reply to point 2: although I follow the comment by the authors that soil input may not be a major source to most lakes, this caveat needs a bit more discussion in the main manuscript so that future readers are not left wondering whether soil input plays a major role. Can some of the reply be added to the main manuscript?

We extended our discussion on the possible effects of soil input on the application of the HDI_{26} to the discussion but kept the discussion brief due to the lack of information on this subject.

Reply letter reply to point 5: so here the authors argue that they only provide a local calibration, but this is nowhere mentioned in the abstract of the paper. In fact, the beginning of the current version of the abstract argues that our understanding of continental climate is hampered by the lack of global applicable proxies and therefore hints at this paper providing a universal proxy. But this paper does not provide a global proxy.

There seems to be a misunderstanding here. We state that the reconstruction of continental climates is hampered by the scarcity of lipid palaeothermometers that are applicable worldwide. Our study provides a calibration for East African lakes and a second more universal calibration and hence we indeed intend to introduce the HDI_{26} as a proxy for the reconstruction of lake water temperatures on a global scale, which is fully supported by the presented data.

To solve this, restructure your abstract (I suggest to delete sentence in line 13-14 (see also below)) and clearly state in the abstract that this is a local calibration for east African lakes only. Maybe even add this local calibration aspect to the title of the manuscript to avoid confusion and blind application of this calibration to other lakes by future studies.

We provide a proxy that is applicable globally as demonstrated by our data set. The calibrations provided in the manuscript allow obtaining past surface water temperatures from lake archives worldwide and they will lie the foundation for future studies that will add to our understanding of the relationship between the HDI_{26} and water temperature on regional to global scales. Understating the potential of the HDI_{26} in palaeoclimate research thus does not seem appropriate here.

Reply letter reply to point 6: I think this is a really good reply and this explanation for a lack of culture data to compare the surface sediment data with, in a condensed form, should be added to the main manuscript.

We are thankful for this comment and now added a brief statement that additional culture and environmental studies are needed in the future to further constrain the relationship of the HDI_{26} with temperature. However, we do not regard it as beneficial to specifically explain why no data from previous studies can be plotted along with the East African and globally distributed lake environments.

Line 13-14: delete this sentence as some universally applicable continental climate proxies exist (MBT_{5Me} to name one, but there are others). See also comment above.

We did not intend to discredit any of the proxies that are currently used to study continental climate change. We here rather referred to the weaknesses and uncertainties with which each proxy is fraught with, making a global application complicated. We modified the sentence and now indicate that there is a "scarcity" of proxies that - on a global scale - allow studying continental climate change.

Line 23: change to: "thereby quantifying past continental climate change."

We changed the sentence according to the reviewer's suggestion.

Line 24: This is not correct. The temperature change (warming) we have so far experienced is at best 1.5 °C since the pre-industrial, even your deglacial temperature record shows more temperature change. I think you might confuse the extent of warming with the rate of warming, which MIGHT be unique?

There seems to be a misunderstanding here. We did not refer to the warming since the industrialisation, which is in the order of ~1 °C as mentioned by the reviewer. Instead, we demonstrate an increase of temperature by 4 °C from the last glacial maximum to the late pre-industrial period. Such a warming is well in line with other proxy reconstructions from this area, showing that the HDI₂₆ allows the reliable reconstruction of past lake surface water temperatures in tropical East African lakes and likely other lakes worldwide.

Line 23-29: this beginning of the abstract still feels forced and not terribly relevant for a proxy calibration paper. I strongly suggest you delete this section and start with explaining the history or (organic) paleoclimate proxies.

This part of the introduction (which is now clearly indicated in the text) provides the narrative for our study. We do feel that embedding our novel proxy in a climate relevant context is more appropriate for the content of the manuscript. We previously provided information on the history of organic temperature proxies (given the word limit) but now extended this part of the introduction by including previously not or only little discussed proxies such as the brGDGT-based MBT_{5Me} and the alkenone-derived U^K₃₇. Therefore, we believe that the suite of possible lipid palaeothermometers for continental research is now sufficiently covered (again within the word limit).

Line 35 (and others): not sure what "punctuated" climate events are. Do you mean abrupt climate change events such as the Younger Dryas?

We indeed refer to short-lived climate change events but understand that the term "punctuated climate events" may be somewhat ambiguous. Therefore, we replaced it (on all occasions) by "abrupt climate change events".

Line 100: can the authors add the altitude of each lake to figure 3? That would help to show the reader that the clustering fits with altitude.

We now included the altitude to each lake in Figure 3.

Line 115-116: delete “widespread and”

We modified the sentence according to the reviewer’s suggestion.

Line 116: When I look at figure 3 it seems that HG₂₆ diols and HG₂₆ keto-ols have extremely low abundances in sample Kn?

Lake Kisibendi (Kn) indeed shows comparatively low relative abundances of HG₂₆ diol-ols and HG₂₆ keto-ols. We assume that the reviewer expects that this might affect the calculation of the HDI₂₆. However, in terms of absolute abundance both HGs occurred in sufficient concentrations and a signal-to-noise ratio that allowed their reliable identification and integration.

Line 120+122: it is not clear to me how these “fractional abundances” were calculated. Can you give the equation for these two fractional abundances here so the reader understands what is discussed and shown in figure 4?

HG₂₆ diols and HG₂₆ keto-ols each occurred in form of two structural isomers in all of the investigated lakes. We calculated their relative proportion as the abundance of each of these components divided by the summed abundances of all HG₂₆ diol and HG₂₆ keto-ol isomers. This is stated in line 118. However, we agree that the calculation of the fractional abundances of HG₂₆ diols and HG₂₆ keto-ols may need additional explanation and included more information on this matter and the relevant equation to the ‘Methods’ section.

Line 141: delete “only”

“Only” was removed from the sentence.

Line 145: This lake has a latitude north of 60 °S, so technically not subpolar I guess but mid-latitudes?

Laguna Potrok Aike is situated at the boundary between the Antarctic Polar Front and the Southern Hemispheric Westerlies and with its position at 51°S it is located within the subpolar climate zone, which is commonly considered to extend from 50 to 70°S. The lake is thus regarded to be subpolar.

Line 190-200: somewhere here the correlation with summer temperature and mean above freezing temperatures needs to be discussed. Why was SWT selected for the proxy? Although the seasonal cycle is small in these African lakes, this should become significant for the high-latitude lakes, as hinted at later in the discussion and as stated in the reply letter (“HG signal captures the signal of highest cyanobacterial in-lake productivity”).

Surface water temperature was selected for the proxy because heterocytous cyanobacteria express buoyancy and accumulate at or close to lake surfaces. The HDI₂₆ thus captures a shallow water temperature signal. This is also corroborated by our statistical analyses, which indicate that the HDI₂₆ shows a stronger correlation with SWT than with deep water temperatures. Therefore, we consider SWT the most appropriate approximation for the reconstruction of lake water temperatures. On multiple occasions throughout the manuscript, we address the seasonality of the proxy and already in the ‘introduction’ section indicate that HGs capture the signal of highest in-lake productivity. In the ‘Discussion’ section, we dedicated an entire paragraph to this topic, in which we elaborate on the effect of seasonality on the HDI₂₆ and explicitly state that the HDI₂₆ does reflect a seasonal and not a mean annual temperature signal in temperate and polar lakes. We thus already provide a

comprehensive discussion of seasonality in the manuscript and in view of the word limit were not able to extend this discussion any further.

Line 243: also give RMSE here

As requested by the reviewer, the RSME has been added to the calibration function.

Line 263-265: how can such community changes that impact the calibration be constrained in the past? How can we assess whether the temperature dependence has changed in down core records due to changes in the cyanobacterial community? This is a major issue that needs to be discussed.

The reviewer addresses an issue that is not only of concern for HG-based calibrations but for all temperature proxies (including e.g. the U^{K}_{37} and the MBT'_{5Me}) and in all of these cases, modern proxy calibrations are employed to reconstruct past temperatures. We completely agree with the reviewer that this a highly sensitive topic but based on our results conclude that changes in the cyanobacterial community are likely to have only a minor impact on the reconstruction of water temperatures using the HDI_{26} . This is based on the observation that in modern East African as well as the globally distributed lake environments, harbouring very different cyanobacterial community compositions, a very strong correlation with SWT is observed. This would not be the case if changes in the overall community composition had affected the calculation of the HDI_{26} . Second, we consciously chose Lake Tanganyika to establish the first continuous HDI_{26} -based temperature reconstruction because this lake has been intensively studied in the past and deglacial temperature reconstructions based on pollen and the TEX_{86} are available to which the HDI_{26} can be compared. The overall good agreement of the HDI_{26} -reconstructed temperature record with that of other climate proxies demonstrates that the HDI_{26} captures long term climate trends as well as abrupt climate change events, despite major shifts in the cyanobacterial community composition as indicated by changes in the HG distribution pattern. Therefore, we conclude that changes in the cyanobacterial community do not overly impact the reconstruction of HDI_{26} -based temperatures but we agree that this is an issue that will have to specifically investigated in future culture and environmental studies.

Figure 5: Can you explain the meaning of the different coloured lines in figure 5? This is currently not clearly stated in the figure legend (e.g. what does the blue line mean, etc).

We used the different colours as optical guidance and to show differences between the various regression lines. We agree with the reviewer that the colour coding needs clarification and now explain the different colours in the figure legend.

Line 305: although maybe not for TEX_{86} , it is (maybe better) constrained for other organic proxies such as U^{K}_{37} and the LDI. Change sentence accordingly.

We now indicate that the habitat depth is an issue that mainly concerns the TEX_{86} .

Line 355-356: something is wrong with this sentence. What does “keeping in with” mean?

In the text, we actually wrote “in keeping with”. This a common expression indicating that a circumstance is “in harmony or conformity with something”. We thus consider the sentence as etymologically and grammatically correct.

Figure 6: This figure needs a bit more work. First, can you add the calibration RMSE for both TEX_{86} and the HDI_{26} calibrations to all the data points to better show the uncertainty associated with each record? Second, what does the error currently shown for HDI_{26} mean, can you state this in the caption? Third: delete the 8.2 kyr event.

We now added the RSME associated with the TEX_{86} and the HDI_{26} calibrations to figure 6. The smaller error indicated for the HDI_{26} relates to the analytical accuracy with which the novel lipid palaeothermometer can be determined based on replicate analyses. The colour coding used to indicate the analytical and calibration uncertainties are now explained in the caption. The 8.2 kyr event has been removed from the figure.

Line 364-375: delete this part. The temperature change during the YD is well within the RMSE of the calibration and based on 1 data point. Just focus on the long-term trends.

We have a slightly different view on this matter. We agree that this interval needs to be investigated in higher resolution in future studies but a decline in temperature during the Younger Dryas has been reported on multiple occasions from tropical East Africa and in terms of timing and magnitude agrees with the change observed in our HDI_{26} record. Therefore, we feel that it is justified to include a brief discussion on the use of the HDI_{26} in studying abrupt climate change events, such as the Younger Dryas.

Line 382: This offset of 1.6 °C is well within the RMSE of both the TEX_{86} and HDI_{26} proxy, so are these two records really statistically different? That is why it would be good to add the RMSE for both these proxies to figure 6.

The observed temperature offset is indeed within the RSME of both proxies. However, we consider it highly unlikely that this offset is an analytical bias. First, it is too consistent in the Holocene sediment sequence and analytical uncertainties are expected to show a more random pattern. Second, the HDI_{26} and the TEX_{86} temperature records show very similar trends in the Pleistocene sediments and it would be surprising if the analytical and calibration errors would be less pronounced here. The offset between both proxies thus seems to be caused by environmental changes in the lake and those are explored in the discussion. However, as requested by the reviewer we added the RSME for both proxies to figure 6.

Data availability: prior to publication the data should also be uploaded to a public database such as Pangaea to secure future access.

All data used in study is submitted along with the manuscript as supplement and if accepted for publication, it will be available to the general public due to the open access policy of Nature Communications. Therefore, the data is secured for future access.